# A 17 year climatology of the macrophysical properties of convection in Darwin

Robert C. Jackson[1], Scott M. Collis[1], Valentin Louf[2], Alain Protat[3], and Leon Majewski[3]

[1]Argonne National Laboratory, 9700 Cass Ave., Lemont, IL, USA
[2]School of Earth, Atmosphere and Environment, Monash University, Clayton, VIC, Australia
[3]Bureau of Meteorology, 700 Collins St, Docklands VIC 3208, Australia

*Correspondence to:* Robert Jackson (rjackson@anl.gov)

**Abstract.**

The validation of convective processes in general circulation models (GCMs) could benefit from the use of large datasets that provide long term climatologies of the spatial statistics of convection. To that regard, echo top heights (ETHs), convective areas, and frequencies of mesoscale convective systems (MCSes) from 17 years of data from C-band POLarization (CPOL) Radar are analyzed in varying phases of the Madden-Julian Oscillation (MJO) and Northern Australian Monsoon in order to provide ample validation statistics for GCM validation. The ETHs calculated using velocity texture and reflectivity provide similar results, showing the ETHs are insensitive to various techniques that can be used. Retrieved ETHs are correlated with those from MTSAT retrieved cloud top heights, showing that the ETHs capture the relative variability in cloud top heights over seasonal scales.

Bimodal distributions of ETH, likely attributable to the cumulus congestus and mature stages of convection, are more commonly observed when the active phase of the MJO is over Australia due to greater mid-level moisture during the active phase of the MJO. The presence of a convectively stable layer at around 5 km altitude over Darwin inhibiting convection past this level can explain the position of the modes at around 2 to 4 km and 7 to 9 km respectively. Larger cells were observed during break conditions compared to monsoon conditions, but only during the inactive phase of the MJO. The spatial distributions show that Hector, a deep convective system that occurs almost daily during the wet season over the Tiwi Islands, and seabreeze convergence lines are likely more common in break conditions. Oceanic mesoscale MCSs are more common during the night over Darwin. Convective areas were generally smaller and MCSes more frequent during active monsoon conditions. In general, the MJO is a greater control of the ETHs in the deep convective mode observed over Darwin, with higher distributions of ETH when the MJO is active over Darwin.

# 1 Introduction

Convection in the tropics has an important impact on the global radiative budget. For example, anvil cirrus that are detrained from convection can have a solar forcing on the order of $100 \ W \ m^{-2}$ (Jensen et al., 1994). The infrared radiative forcing of these anvil cirrus is highly dependent on the temperature, or height where they are present. Furthermore, convection acts as a vehicle to transport moisture to the tropical tropopause layer (15 km) (TTL) (Dessler, 2002) and therefore can significantly affect the distribution of moisture at the tropopause. Furthermore, the largest convective cells account for a majority of the convective mass flux (Kumar et al., 2015; Masunaga and Luo, 2016; Hagos et al., 2018) This shows that the convective cell size is another important factor for determining vertical moisture transport. Therefore, knowledge on the cloud top heights and cell sizes of such convection is useful for determining the impact of deep convection on the global radiative budget and upper tropospheric distribution of moisture. This provides a need for a climatology of such cloud macrophysical properties in the tropics that can be used to validate global climate model (GCM) simulations of convection.

A region in the tropics with continuous observations of the macrophysical properties of clouds provides such a climatology. Such a region is located in Darwin, Australia. Decades of continuous observations have been collected in Darwin, including 17 years of plan position indicator (PPI) scans from the C-band Polarization Radar (CPOL). This region is also ideal for developing such a climatology as the synoptic scale forcing can be objectively determined from both determining information about the Northern Australian Monsoon (Drosdowsky, 1996; Pope et al., 2009) and about the phase of the MJO (Madden and Julian, 1971; Wheeler and Hendon, 2004), both of which can provide the forcing necessary for convection to develop over the Darwin area. For example, high (20-30 dBZ) reflectivities above the freezing level, slightly smaller cells, and greater amounts of lightning have been observed during break convection while monsoonal convection tends to be shallower, more widespread, and less electrically active (Rutledge et al., 1992; Williams et al., 1992; May and Ballinger, 2007; Kumar et al., 2013a, b). Also, the monsoon preferentially onsets when the convectively active phase of the MJO approaches Darwin (Evans et al., 2014), so the MJO and the monsoon are not necessarily independent of each other. Besides convection associated with the MJO and the monsoon, the convection that occurs in this region can also be influenced by the seabreeze and Tiwi Islands, such as the nearly daily summer occurrence of an intense deep convective system called Hector (Keenan et al., 1989; Crook, 2001). The MJO is poorly resolved in many general circulation models (GCMs) (Gu et al., 2011). Also, convective parameterizations in GCMs do not account for mesoscale organization resulting in insufficient sensitivity to upper tropospheric humidity (Del Genio, 2012). Therefore, statistical analyses of how the cloud tops vary for differing phases of the MJO and monsoon are useful for GCM validation.

Past studies have examined the cloud top heights in convection over Darwin and the maritime continent estimated by radar through the echo top height (ETH). Observations in Indonesia by Johnson et al. (1999) showed 3 modes in ETH in convection during the Tropical Ocean - Global Atmosphere Coupled Ocean Atmospheric Response experiment that corresponded to stable layers: a mode at 2 km, a mode at around 5 km, and a mode at around the tropopause of 15 km. Over Darwin, May and Ballinger (2007) examined the distribution of ETHs in convection for one wet season in and found limited evidence of multimodal distributions of ETH, but had only considered the maximum cloud top height over a cell's lifetime. Furthermore,

they found that cloud top heights in break convection were higher than those in monsoonal convection. Kumar et al. (2013b) analyzed two wet seasons of CPOL data and found evidence of bimodal ETH distributions. They also found that convection formed during active monsoon conditions has lower cloud top heights than convection formed during break conditions similar to May and Ballinger (2007). Kumar et al. (2013a) investigated 3 wet seasons of CPOL data in Darwin and found four differing

modes that corresponded to trade wind cumulus, cumulus congestus, deep convection and overshooting convection. Therefore, differing conclusions have been reached on the number of modes of convection that are present over Darwin and the maritime continent.

This study improves upon past studies looking at cloud top heights in Darwin in various ways. First, the analysis is expanded to the full CPOL record of 17 wet seasons to analyze the relationship between convective properties and the large-scale

environment on a more statistically representative dataset than has been done in Johnson et al. (1999); May and Ballinger (2007); Kumar et al. (2013a, b). This is possible using recent advances in supercomputing and recent developments of highly customizable distributed data analysis packages written in Python such as Dask (Dask Development Team, 2016). Secondly, none of the past studies have looked at how ETHs can vary for differing phases of the MJO. Rauniyar and Walsh (2016) have found that rainfall rates in Darwin are correlated to the presence of the convective phase of the MJO over Darwin. Also,

Evans et al. (2014) have observed that the preferential onset of the monsoon is when the active phase of the MJO approaches Australia. Therefore it is worth exploring the possibility that cloud top heights of convection in Darwin are also influenced by the MJO. In this study we wish to answer the following questions:

1. Does the MJO have an influence on the observed convective cloud top heights over the Darwin area?

2. Do the conclusions of past studies regarding the heights of break and monsoonal convection over Darwin still hold when

using 17 wet seasons?

3. What are the spatial distributions and diurnal cycles of convective cloud top heights over Darwin during the differing phases of the MJO and monsoon?

4. When do we observe multiple modes in the distributions of cloud top heights when looking at 17 years worth of data?

It is also important to note that the past studies have used a reflectivity (Z) threshold to determine the ETH. Using a reflectivity

threshold, there is the potential that regions near cloud top that are detected by CPOL could be excluded. Reflectivity fields from CPOL are also prone to miscalibration and attenuation, although these effects have been corrected for as best as possible in the dataset used in what follows. As an independent assessment of these corrections and for sake of comparison of two ETH retrieval techniques, this study also assesses the applicability of an alternate algorithm that uses velocity texture $\sigma$, a quantity not affected by radar miscalibration and less likely to be affected by attenuation than Z, to derive the ETH. This technique also

allows for automatic detection of the noise floor of the radar and potentially the inclusion of more regions near cloud top. The feasibility of using this methodology is assessed by comparing ETHs derived using varying thresholds of $\sigma$ as well as as Z against satellite retrieved cloud top heights.

The remainder of the paper is organized as follows. Section 2 will go into detail about the data products that are used. Section 3 describes the algorithms used to process the radar data, estimate echo top height, and quantify the synoptic scale forcing over

Darwin. Section 4 shows results on how the ETHs vary in differing phases of the MJO, differing monsoonal regimes. Section

also shows the diurnal cycle of ETHs over Darwin over varying regions and in various phases of the MJO and monsoon in order to link the ETHs to both large scale and localized mechanisms. Section 5 shows the primary conclusions of this study.

## 2 Data products

### 2.1 CPOL

The C-band Polarization (CPOL) radar, located at the Tropical Western Pacific (TWP) Atmospheric Radiation Measurement site in Darwin, Australia (Keenan et al., 1998), conducted Plan Position Indicator (PPI) scans every 10 minutes at 15 elevations ranging from $0.5°$ to $40°$ with a 300 m resolution and gate spacing during the summer seasons of 1998 to 2017, excluding 2007 and 2008. While CPOL records polarimetric variables, the variables of interest from CPOL for retrieving ETH are the reflectivity $Z$ and the radial velocity $v_r$. The reflectivity $Z$ was calibrated using a technique that combines information from

ground clutter and spaceborne radars that calibrated $Z$ from CPOL within 1 dB (Louf et al., 2018). Other instrumentation was used to supplement information provided by CPOL. Rawinsonde launches were conducted four times per day from the ARM Facility TWP site, providing vertical profiles of temperature T, dew point $T_d$, relative humidity RH, and the zonal ($u$) and meridional ($v$) components of the wind velocity. In order to estimate the uncertainty in the retrieved ETHs from CPOL, we compare the ETHs against satellite retrieved cloud top heights (CTHs). To do this, we use the Japanese Multi-functional

Transport Satellites (MTSAT) which images brightness temperature $T_b$ every hour at a 4 km resolution, giving information on CTHs estimated by the Visible Infrared Solar-Infrared Split Window (VISST) Technique during the day and the Solar-Infrared Infrared Split Window (SIST) Technique during the night (Minnis et al., 2011). The MTSAT are integrated into the VISST data product version 4 (Minnis et al., 2011) providing cloud top height over a $12°$ by $12°$ region centered over Darwin.

## 3 Data processing algorithms

### 3.1 Radar data processing

The previous section gave information about the radars and other instruments used in this study. This section details how the echo top heights were estimated from CPOL and evaluated against satellite measurements. The Python ARM Radar Toolkit (Py-ART) (Helmus and Collis, 2016) was used in order to process, grid, and display $v_r$ and Z. Before the velocity texture was calculated, gates that met any of the following criteria were removed in order to remove ground clutter and noise:

1. Z < -20 dBZ or Z > 80 dBZ

    2. differential reflectivity > 7 dB or < -3 dB

    3. Texture of differential phase greater than $20°$

    4. Cross-correlation coefficient < 0.45

The velocity texture was then calculated using the standard deviation of the 3 x 3 window surrounding a gate and then was gridded onto a Cartesian grid at a 1 km horizontal and 0.5 km vertical resolution using Barnes (1964)'s interpolation technique. Since at higher elevations and ranges some gaps in the interpolated radar field may be present due to lack of radar sampling at a given location, a radius of influence that increases as a function of distance from the radar was used in order to minimize artifacts generated by radar sampling. Due to the increasing beam width as a function of distance from the radar as well as decreased vertical resolution with height, there is greater uncertainty in the the ETH as distance increases from the radar. Therefore, only data on a 200 by 200 km box surrounding CPOL was used, following (Kumar et al., 2013b) and data less than 20 km from the radar was excluded as the radar does not scan at heights of 20 km at these distances. We use Dask (Dask Development Team, 2016), a package written in Python to analyze datasets on distributed clusters, to map the problem of analyzing the 17 years of CPOL data to the Bebop cluster at Argonne National Laboratory.

### 3.2 Calculation of ETHs

Now that the radar data has been processed and interpolated onto a Cartesian grid, the next step is to calculate the ETH from CPOL. Past studies (May and Ballinger, 2007; Kumar et al., 2013a, b) have used the highest pixel in the column with Z > 5 dBZ as the ETH, while only including time periods where there were echoes present in all pixels from 2.5 km to the highest pixel with Z > 5 dBZ to remove time periods with a cloud layer above the precipitating convection. Since this threshold can be at least 2 dB above the minimum detectable signal at a range within 100 km of CPOL as seen in Figure 1, using reflectivity (Z) could potentially remove regions that are both detected by CPOL and in cloud, especially for clouds with heights < 10 km and ranges < 60 km from CPOL. Also, Z is prone to errors from both miscalibration and attenuation in heavy rain. Therefore, we examine the possibility in this study of using Doppler velocity texture $\sigma$ as a threshold instead of $Z$, which is immune from errors from miscalibration and less prone to errors from attenuation than $Z$.

Figure 2 shows and example of how $\sigma$ is interpreted. To demonstrate that higher values of $\sigma$ correspond to noise, Figs. 2a,b show an example field of $Z$ and $\sigma$ from CPOL for a case of isolated convection on 05 March 2006. Figure 2b shows isolated regions of $\sigma < 3$, corresponding to the regions of precipitation that are seen in Figure 2b. The more widespread regions of $\sigma > 3$ correspond to clutter, noise, and multi-trip echoes that are present in Fig. 2a. This is due to the fact, that, in a phase randomized radar such as CPOL, regions of significant returns will have smoother velocity fields and regions of noise will have more random velocity fields, allowing for the automatic detection of the minimum detectable signal. When a threshold of $\sigma > 3$ is used to mask gates in Fig. 2a, Fig. 2c shows that only regions of precipitation are still present after masking, showing that noise is removed. The ETHs are then determined by looking at the lowest gate in the column that is masked. We use the lowest gate in the column in order to ensure that we are capturing the ETHs of the precipitating convection, and not that of detrained anvils and cirrus that can lie above the precipitating convection.

However, even despite using a methodology that automatically determines the minimum detectable signal using $\sigma$ that is immune to radar miscalibration and less prone to errors from attenuation, there is still the possibility that the smallest cloud water droplets near cloud top are not detected by CPOL. Therefore, it is important to assess quantitatively whether the ETHs calculated using the $\sigma$ threshold represent the variability in cloud top heights that is observed in convection in Darwin and to

test the sensitivity of the retrieved ETHs to the retrieval technique used. To do this, ETHs from CPOL are compared against CTHs retrieved by MTSAT which provide an independent estimate of CTH. Since the MTSAT data are at an hourly temporal resolution, only CPOL scans that were within 10 minutes of a MTSAT record were compared against the MTSAT retrieved cloud top heights from March 2006 until December 2010. This time period was chosen as it is the time period covered by

Version 4 of the VISST product over Darwin and ensures that the same data processing techniques were used throughout the dataset. Since the two datasets are at differing resolutions, the MTSAT data are interpolated onto the same grid as the CPOL data for the comparison. Furthermore, to ensure that we are comparing points that are in precipitating convection we both only include points from MTSAT where the VISST product identified cloud and where the convective classification algorithm, detailed in Section 3.4, classified the grid points as convection.

Figure 3 shows the results of such comparisons using the lowest gate in the column where Z < 5 dBZ, with the 5 dBZ threshold from May and Ballinger (2007) and Kumar et al. (2013a) (Figure 3a). Figures 3bcd use varying thresholds of $\sigma$ for the ETHs. In all panels of Figure 3, there is considerable spread in the comparison between the satellite CTHs and the ETHs retrieved by CPOL. Time periods within two hours of sunrise and sunset were excluded in the generation of Figure 3 since large uncertainties can exist in the VISST retrieval during twilight hours. Since VISST uses two different retrieval algorithms

for day and night time, this can potentially introduce large uncertainties in the retrieved CTHs from MTSAT. However, when the data were stratified by day or night time, there was little difference in the generation of Figure 3 (not shown). The coarser resolution of the MTSAT data compared to the CPOL data indicates that smaller scale heterogeneity may not be captured by MTSAT. Nevertheless, it is expected that, over time scales of years and spatial scales of 100 km, the MTSAT captures the interseasonal variability in CTH.

There is a clear peak in the frequency distribution present in all panels at ETHs > 7.5 km and the median ETH is within 4 km of the VISST retrieved cloud top height for ETHs > 2.5 km. Furthermore, the average ETH retrieved from CPOL increases with increasing MTSAT retrieved cloud top height in Figure 3, showing that the ETHs retrieved from CPOL are correlated (Pearson correlation coefficient of 0.49, p < 0.01 according to a $\chi^2$-test) with the satellite retrieved cloud top heights. As cloud top heights retrieved by MTSAT can have an uncertainty as high as 3 km (Hamada and Nishi, 2010), it is not possible to

determine whether the MTSAT retrieved cloud top heights or CPOL ETHs provide the better estimate of cloud top height. Nevertheless, the correlation between the CPOL ETHs and satellite retrieved cloud top heights shows that the CPOL retrieved ETHs using any of the tested techniques capture the statistical variability in cloud top heights that are observed in Darwin. The mean, and fifth and ninety-fifth percentiles are similar in Figure 3abcd, showing that there is little difference between using May and Ballinger (2007) and Kumar et al. (2013a)'s technique compared to using $\sigma$ for calculating ETH. Therefore, there is

little sensitivity of the retrieved ETH to the technique used, showing the robustness of the ETH retrieval. Therefore, this study uses the lowest gate in the column where $\sigma > 3$ to be defined as the ETH. Because of this definition, the fact that, when the VISST CTH is greater than 10 km, the median CPOL ETH is 4 to 5 km lower than the VISST retrieved CTH in Figure 3 shows that there are commonly multiple layers of cloud present. This therefore shows that CPOL is able to detect the presence of multiple cloud layers while the VISST technique can only detect the highest cloud layer and shows an advantage of using the

CPOL ETHs over the VISST CTHs.

## 3.3 Quantification of large scale forcing

Now that a methodology has been developed to calculate the echo top height from the CPOL data, the next step in this study is to develop methodologies for quantifying the large scale forcing in the Darwin region. The large scale forcing in the Darwin region can be quantified with respect to two major synoptic phenomena. One of them is the Northern Australian Monsoon (Drosdowsky, 1996; Pope et al., 2009). Its presence is characterized by deep westerly winds over Darwin that provide moisture flow from the Indian Ocean to the Darwin region. Many algorithms have been used in the literature to determine the presence of the Northern Australian Monsoon. However, only Drosdowsky (1996) and Pope et al. (2009) robustly identify the presence of the monsoon. These algorithms depend on the profile of the zonal component $u$ and meridional component $v$ of the wind as well as temperature, dew point, and pressure collected by rawinsondes over Darwin. The first, Drosdowsky (1996), uses the deep-layer (Surface-500 hPa) mean $u$ in order to characterize the presence of the monsoon. Under this classification, a deep layer of westerly winds is characteristic of the monsoon, which provides an environment where moisture is flowing from the Indian Ocean to Darwin. The second, Pope et al. (2009), uses $k$-means clustering on the winds, temperature, and dew point to find five regimes that correspond to differing synoptic scale phenomena in Darwin: deep west, moist east, east, dry east, and shallow west. The "deep west" and "shallow west" regimes corresponds to westerly flow from the surface to 500 hPa, or the active monsoon. The "moist east" regime corresponds to (Drosdowsky, 1996)'s "break" regime where continental convection is more likely to occur, and the other regimes correspond to either suppressed or transitional regimes.

The large scale forcing can also be quantified by the phase of the Madden-Julian Oscillation (Madden and Julian, 1971). The phase of the MJO is quantified using a number 1 to 8 that gives an indicator of the position of the enhanced and suppressed convective activity associated with the MJO. When the MJO phase increases from 1 to 3, the enhanced convective activity is traveling to the east from the Indian Ocean. When the MJO index increases from 4 to 7, the enhanced convective activity is over the maritime continent and traveling east to the Pacific Ocean. When the MJO index is 8, the enhanced convective activity is over the Pacific Ocean. This MJO index is determined using Wheeler and Hendon (2004)'s database that is based on both the outgoing long-wave radiation from satellites and the 850 to 200 hPa $u$ from reanalysis data. Rauniyar and Walsh (2016) found that the yearly occurrence of the Pope et al. (2009) regimes has a high amount of interannual variability as they are modulated by the El Nino-Southern Oscillation (ENSO) while the occurrence of the MJO does not. Therefore, the use of Pope et al. (2009) is less suitable for a sample size of 17 seasons than the use of the MJO for analyzing such a dataset. Furthermore, the five classifications provide more opportunity for over-classification than using Drosdowsky (1996). Therefore, to quantify the large scale forcing over Darwin, the dataset is separated using only the MJO index and the Drosdowsky (1996) monsoon classification. To even further prevent the possibility of over-classification, much of the data in this study are classified into whether the convective phase of the MJO is over Australia (MJO indicies 4 to 7), or when it is not (MJO indicies 1 to 3, 8).

Figures 4 and 5 show the mean thermodynamic and wind profiles for given phases of the MJO and monsoon. Figs. 4cd show that break conditions are generally characterized by a layer of east-northeasterly winds extending to about 10 km when the MJO is over Australia and throughout the troposphere when the convective mode of the MJO is elsewhere, while westerlies are prevalent at altitudes up to 8 km during monsoon conditions. Figure 4b shows lower fifth percentiles of relative humidity

at 4 to 8 km than Figure 5b, demonstrating a greater number of cases with drier midlevels during break conditions. This is suggestive of conditions that would support deep convection would be less prevalent during active break conditions than during the monsoon (Hagos et al., 2013).

There is a greater variability in the winds below 6 km when the active phase of the MJO is over Australia in Figures 4cd and 5cd. In particular, the 95th percentiles of $u$ are greater below 6 km when the MJO is active over Australia than when the active phase of the MJO is away from Australia in in Figures 4c and 5c. This could lead to enhanced advection of moisture from the Indian Ocean over Darwin. To examine whether this is the case, the moisture flux, calculated as $(u^2 + v^2)^{1/2}q$ where $q$ is the water vapor mixing ratio, was estimated from the rawinsondes and plotted into Figure 6 for each of the conditions in Figures 4 and 5. Figure 6 shows that, in the lowest 1 km of the atmosphere, there are greater moisture fluxes during and active MJO and and active monsoon, showing that there is greater advection of moisture over Darwin during both an active MJO and an active monsoon. The enhancement of moisture advection associated with the monsoon is stronger than that associated with the MJO. Furthermore, the difference between the ninety-fifth and fifth percentiles of relative humidity at 4 to 8 km are greater when the active phase of the MJO is away from Australia, as seen in Figs. 4cd and 5cd, showing more variability in mid-level moisture when the MJO is inactive over Australia. Mean relative humidities are 3 to 20% higher when the MJO is active over Australia, as seen in Figures 4b and 5b. Therefore, conditions are expected to be consistently more favorable for supporting deep convection during the active phase of the MJO (Hagos et al., 2013).

The thermodynamic profiles in Figures 4a and 5a then show a transition from convectively unstable conditions from 1 km until 4 km to stable at around 4-6 km, or temperatures around $0°C$ with the tropopause located at around 15 km. Three stable layers over Indonesia were observed by Johnson et al. (1999): a trade wind layer at 2 km, a stable layer at around 6 km, and the tropopause at 15 km. This suggests that, while the trade wind stable layer is not present over Darwin, the stable layer at around around $0°C$ that is present over Indonesia is also present over Darwin. The presence of such a layer suggests that any parcel lifted from the surface would easily rise to heights to around 4-6 km, or $0°C$ but would need to be dynamically energetic enough to penetrate through the stable layer above this level. Furthermore, since the stable layer is located at temperatures just below $0°C$, this inhibits the formation of ice which releases latent heat that would invigorate the updraft, further inhibiting convection. Therefore, an updraft strong enough to reach levels where ice formation to occur would be more likely able to penetrate up to the tropopause. This notion is supported by the observations of vertical velocities in convection that reaches the tropopause typically being strongest above 5 km (Cifelli and Rutledge, 1994, 1998; May and Rajopadhyaya, 1999; Collis et al., 2013; Varble et al., 2014). This therefore indicates that the thermodynamic profiles favor the formation of two distinct populations of convection: weaker cumulus congestus that do not penetrate heights much past the stable layer and stronger deep convection that would more than likely be able to penetrate the tropopause.

### 3.4 Quantification of convective areas and MCSes

In addition to the ETH, two other macrophysical properties of the convective systems sampled by CPOL were derived in this study. One, the convective area, was calculated by taking the total area of continuous regions in each scan that were identified as convective by the algorithm of Steiner et al. (1995). This algorithm defines regions with $Z > 40$ dBZ as well as examines the

peakedness of the $Z$ field in order to classify regions as convective. This algorithm was designed for convection in the Tropics, and a visual analysis of the algorithm showed that it reasonably distinguished between stratiform and convective regions (not shown). The number of MCSes per radar scan was also determined. This was done by using the methodology previously used by Rowe and Houze (2014), Nesbitt et al. (2006), and others which first defines precipitation features (PFs) as all continuous

regions where $Z > 15$ dBZ. After that, the minimum size ellipse that fits around the PF is calculated using the algorithm in Bradski (2000). MCSes are then identified as any ellipses that have a major axis length of greater than 100 km. The number of MCSes were then normalized by the number of scans that were identified to be in the large scale forcing regimes in Section 3.3 and the calculations placed into Table 1. This was done in order to account for the differences in the amount of time spent in each large scale forcing regime over the study period.

## 4   Statistical analysis/discussion

### 4.1   Normalized frequency distributions of ETH and convective area

The previous section detailed the methods and uncertainties in the derived ETHs and gave a meteorological overview of the differing synoptic scale regimes. In this section, the entire 17 year record of ETHs in differing synoptic scale regimes is analyzed in order to provide a climatology of convective cloud top heights. In this analysis, only convective regions are

considered, using the Steiner et al. (1995) as mentioned in Section 3.4 to define convective regions.

Figure 7 shows the normalized frequency distributions of ETHs in differing MJO indicies and monsoonal classifications. Some of the distributions shown in Figure 7 are bimodal with one mode at an ETH of approximately 4 to 8 km and the other at about 9 to 13 km. The normalized frequency distributions in Figures 7efg provide the best examples of such bimodal normalized frequency distributions. Two of the modes are therefore similar to the largest two modes observed by Johnson et al. (1999):

a mode at approximately 4 to 6 km corresponding to the cumulus congestus stage and a mode at 8 to 10 km corresponding to deep convection. Kumar et al. (2013a) observed 4 modes, with the trade wind cumulus mode at 2 km, congestus mode at heights of 3-6.5 km, deep convection mode at 6.5 to 15 km, and overshooting convection at heights greater than 15 km. Some evidence of overshooting convection is present in 7, particularly during break conditions in 7a-h. Also, some evidence of the trade wind mode is visible in Figures 7a-h. However, since the 2 km modes in Johnson et al. (1999); Kumar et al. (2013a)

were observed using measurements with a cloud radar that would be more sensitive to liquid cloud droplets than CPOL, more sufficient quantification of this mode would require a radar with a lower minimum discernable signal than CPOL.

Therefore, we generally observed bimodal distributions of ETH. May and Ballinger (2007) observed limited evidence of bimodality, but bimodal distributions of ETH were also found by Kumar et al. (2013b), corresponding to similar heights. The thermodynamic profiles shown in the previous section can explain the bimodality seen in Figure 7. In particular, the presence

of a stable layer at heights above 5 km can explain the bimodality seen in some of the regimes in Figure 7. As convection develops and evolves into deeper convection, it will first have to grow to the height of the start of the stable layer and will stop growing in vertical extent there if there is not enough buoyancy to penetrate past it. Updrafts would have to be energetic enough to penetrate the cap, and such updrafts would more than likely penetrate up to the tropopause.

In order to quantify the locations and contributions of the peaks in each p.d.f in Figure 7, bimodal Gaussian fits of the form in Equation (1)

$$P(x) = A \frac{1}{\sigma_1 \sqrt{2\pi}} e^{-(x-\mu_1)^2/2\sigma_1^2} + (1-A) \frac{1}{\sigma_2 \sqrt{2\pi}} e^{-(x-\mu_2)^2/2\sigma_2^2} \qquad (1)$$

where $A$ is the contribution of mode 1, $\mu_1$, $\mu_2$ are the mean and $\sigma_1$ and $\sigma_2$ are the standard deviations of modes 1 and
2 respectively were generated for each regime. Mode 1 is the mode with smaller ETHs and mode 2 is the mode with larger ETHs. In Figure 8, $\mu_1$ ranges between approximately 2 and 6 km and $\mu_2$ ranges between approximately 7 and 10 km when the distributions are bimodal ($0.1 < A < 0.9$). This shows that, when the distributions are bimodal, the location mode 1 roughly corresponds to the location of the cumulus congestus mode observed by Johnson et al. (1999); Kumar et al. (2013b), and the location of mode 2 corresponding to their deep convective mode but $\mu_1$ was sometimes greater than 6.5, when A was less than
0.1, indicating a unimodal distribution that corresponds to Kumar et al. (2013b)'s deep convective mode. Therefore, in order to properly identify what types of clouds were present, thresholds based on the bimodality and the locations of the modes are required. Therefore, for the rest of this paper, we define the congestus and deep convective modes as follows:

1. If the ETH distribution is bimodal ($0.1 < A < 0.9$) then mode 1 is the congestus mode and, mode 2 is the deep convective mode

2. If the ETH distribution is unimodal ($A < 0.1$) and $\mu_2 < 6.5$ km then the single mode is the congestus mode, otherwise the single mode is the deep convective mode.

3. If the ETH distribution is unimodal ($A > 0.9$) and $\mu_1 < 6.5$ km then the single mode is the congestus mode, otherwise the single mode is the deep convective mode.

The 6.5 km threshold was chosen following the threshold between congestus and deep convection used by Kumar et al. (2013b).
Following this, from the $\mu_1$, $\mu_2$, and $A$ if one mode was identified using criteria 2 or 3 listed above, then the location of the congestus (deep) mode is $\mu_2$ if criteria (2) was used to identify the single mode and the location is $\mu_1$ if criteria (3) was used to identify the single mode. Since $\mu_1 < 6.5$ km $< \mu_2$ when bimodal distributions were identified, the contribution of the congestus mode was defined to be $A$ and $1-A$ for the deep convective mode. In addition, for the bimodal distributions, $\mu_1$ is the location of the congestus mode and $\mu_2$ the location of the deep convective mode. The locations and contributions of each mode from
this classification were then plotted onto Figure 8.

The contributions of the two modes in Figures 7 and 8 vary with MJO index. As the active phase of the MJO is over Australia (MJO indicies 4 to 7), in Figure 8b the contribution of the congestus and deep convective modes are 40 to 60% indicating bimodal distributions. However, the frequency distributions are less bimodal when the active phase of the MJO is away from Australia as shown in Figures 8b,d. Between MJO phases 1 and 2 in Figure 8d, there is a switch from a unimodal
distribution of congestus to a unimodal distribution of deep convection. However, given that the location of the mode is 7.5 km during MJO phase 1 and 6.5 km during MJO phase 2, this apparent switch is likely attributable to the threshold chosen to

define deep convection and shows a limitation in determining convection type by ETH alone. In active monsoon conditions, more unimodal distributions with lower ETHs are observed as shown in 8b,d when the active phase of the MJO is away from Australia. The enhanced mid-level specific humidities during active monsoon/MJO phases in Figures 4d and 5d provide an environment that supports the transition of congestus to deep convection and is likely contributing to the enhanced bimodality
of ETHs observed when the active phase of the MJO is over Australia (Hagos et al., 2013).

The locations of these modes also vary depending on whether or not the active phase of the MJO is present over Australia. In Figure 8a, on average the cumulus congestus mode is located at approximately 3.0 (2.8 km) when the MJO is active (inactive) over Australia during break conditions in Figure 8a. During monsoon conditions Figure 8b, this mode is on average at approximately 4.4 km (6.2 km) when the MJO is active (inactive) over Australia. The average locations of the deep convective
mode during break conditions are approximately 8.1 and 7.9 km during active and inactive MJO conditions respectively. For the monsoon, the average locations are approximately 8.9 km and 7.5 km during active and inactive MJO conditions respectively. However, a greater percentage of the ETHs are greater than 14 km in Figure 8a than in Fig. 8c, showing that more overshooting tops are present during break conditions. Therefore, on average, when the MJO is active over Australia with more overshooting tops present during break conditions.

The change in heights of the deep convective mode associated with an active MJO over Australia is greater than that associated with the monsoon, while congestus are more sensitive to the presence of the monsoon. Considering that, in Figures 4a and 5a show greater increases in equivalent potential temperature with height above the 5 km stable layer during MJO inactive conditions, this suggests that the midlevel thermodynamic profiles support greater inhibition of the convection in the deep convective mode when the active phase of the MJO is away from Australia. Furthermore, Figures 4cd and 5cd showed enhanced
mid-level moisture when the MJO was active over Australia which creates an environment more favorable for supporting deep convection (Hagos et al., 2013). Evans et al. (2014) noted that the monsoon over Darwin will preferentially onset at MJO indicies 3 and 4, so that also suggests that future studies that examine the properties of convection in Darwin in differing large scale conditions must also consider the phase of the MJO in addition to the monsoon, as the MJO has been shown here to be of importance.

The convective area is another important indicator of organized convection, as larger convective areas are typically associated with stronger updrafts and more organized convection. Therefore, the normalized frequency distribution of the convective areas are also plotted in Figure 9 for differing phases of the MJO, monsoon, and time of day. During the day, we see similar distributions of convective area between break and active monsoon conditions in 9a when the MJO is active over Australia. However, when the active phase of the MJO is away from Australia, the normalized frequency distribution narrows in active
monsoon conditions, showing a fewer cells with areas greater than 100 $km^2$ and more cells with areas in between 7 and 20 $km^2$. At night, the difference in the distribution of convective areas between break and monsoon conditions in Figure 9b is greater than that in Figure 9a, with broader distributions of convective areas during break conditions. In general, little trend in the normalized frequency distribution is seen with changing MJO index in Figure 9. This is indicative of stronger updrafts during break conditions compared to monsoon conditions, especially at night and during inactive MJO conditions. While there is more
widespread coverage of MCSes during active monsoon and when the active phase of the MJO is over Australia as indicated in

Table 1, the analysis in Figure 9 shows that the lower convective areas observed in active monsoon conditions indicate weaker, but more frequent MCSes. Meanwhile, the fewer MCSes observed, larger cell areas, and the increased presence of overshooting tops in Figure 8a shows that less widespread, but stronger convection is present during break conditions, but only at night and when the active phase of the MJO is away from Australia. This is consistent with what has been observed in studies such as Rutledge et al. (1992), Williams et al. (1992), May and Rajopadhyaya (1999), and May and Ballinger (2007).

## 4.2  Diurnal cycle and spatial distribution of ETHs

The previous section established that the MJO is a significant control of the ETHs observed over Darwin. However, the mechanisms by which convection over Darwin can be generated can not only depend on the large scale forcing but can also be influenced by localized mechanisms such as seabreezes. Therefore, to investigate under what conditions the formation of convection via localized mechanisms is more likely, Figures 10 and 11 show normalized frequencies of occurrence of ETHs > 7 and < 7 km for the given synoptic scale forcing shown in Figures 4 and 5 as a function of space and time. The threshold of 7 km is shown as it is, on average, the local minimum between the cumulus congestus and deep convective modes in Figure 7. Figures 10ab show that, in break conditions, the cumulus congestus are confined to Tiwi islands and the Australian continent during the day and more confined to the ocean at night in Figure 10ef. When the MJO is active over Australia, greater counts are present over the ocean during the day in Figure 10a compared to b than when the active phase is away from Australia. Similar conclusions can be made for deep convection during break conditions in Figures 10cdgh.

The peak in deep convection isolated over the Tiwi islands present during the day in Figure 10abcd is a deep convective system that forms almost daily during the wet season called Hector the Convector (Keenan et al., 1989; Crook, 2001). It is likely forced by seabreeze convergence lines and further intensified by cold pools that formed from neighboring cumulus congestus (Dauhut et al., 2016). Given this, in break conditions, both surface level easterly flow onto the Tiwi Islands providing a seabreeze and the presence of cumulus congestus over the Tiwi Islands in Figure 10ab show that the environment is favorable for Hector to occur. The reduced widespread cloud cover that is observed during break conditions (May et al., 2012) as well as MCS coverage (Table 1 also provides an environment more favorable for localized seabreezes to develop around the Tiwi Islands, and hence for Hector to form. A maximum in rainfall over the Tiwi Islands, attributed to Hector, when the MJO is inactive over Australia has been noted by Rauniyar and Walsh (2016). Figure 10d shows a greater frequency of deep convective ETHs over the Tiwi Islands when the MJO is inactive over Australia during the day, which is consistent with increased rainfall over this region. During the night, both congestus and deep convection are more focused towards the oceans and the northern Australian coast in Figure 10efgh. An overnight peak in rainfall in the tropics has been attributed to the presence of long lived oceanic MCSs by Nesbitt and Zipser (2003). The data in Table 1 suggests that there is a greater coverage of MCSes during the night during break conditions, especially when the MJO is active over Australia. Kumar et al. (2013a) showed a peak in ETHs over the ocean in the early morning hours and over the coast and continents during the afternoon, so this study extends their conclusion to 17 wet seasons of data.

In active monsoon conditions, cumulus congestus are widespread throughout the western half of the region in Figure 11a,e while the deeper convection is mostly focused on the Australian coast in Figure 11ce. Fewer occurrences of deep convection

on the Tiwi islands are present in in Figure 11cd compared to break conditions, suggesting that conditions are less favorable for the formation of Hector during an active monsoon. Since westerly winds at the surface to 500 hPa are prevalent during an active monsoon, the seabreeze is most likely to occur over the western coast of the Tiwi islands and the western Australian coast over Darwin and therefore any seabreeze convergence lines that do form would form there. However, as active monsoon conditions are typically characterized by widespread stratiform cloud cover (May et al., 2012), and there are a greater number of MCSes in active monsoon conditions (Table 1), it is less likely that diurnal solar heating would be as important during an active monsoon. The reduced solar heating during an active monsoon is likely resulting in the reduced occurrences of Hector, as the lack of solar heating decreases the likelihood that localized seabreezes form.

At night, both congestus and deep convection decrease in occurrence on the continents as one goes east in Figures 11e-h. This suggests that oceanic MCSs, common during an active monsoon as suggested by May et al. (2012) and Table 1, are decaying as they approach land from the west as would be expected with the deep westerly flow. During the night they would encounter a drier airmass than over the ocean, causing a depletion of moisture and hence decay. Figure 11 also shows more sporadic occurrences of both congestus and deep convective clouds when the MJO is inactive over Australia (Figures 11f,h) compared to when it is active (Figures 11e,g). There are 63 days where the MJO was both inactive over Australia and where an active monsoon occurs, while there are 54 days where the monsoon and MJO are both active over Australia. Meanwhile, $7.9 \times 10^5$ occurrences of ETH > 7 km are present when the MJO is inactive and $2.5 \times 10^6$ are when the MJO is active in Figure 11. Therefore, the more sporadic occurrences are not due to the fact that the monsoon and MJO were active for fewer days, but rather this suggests that, even in the presence of deep westerlies that are characteristic of the monsoon, convection is suppressed when the MJO is inactive over Australia. Indeed, Evans et al. (2014) have found that the onset of the monsoon preferentially starts when the MJO indicies are 3 and 4 and decays when MJO indicies are 7 and 8 as the convective phase travels east over the maritime continent. Therefore, it is not surprising to see fewer occurrences of convection when the MJO is inactive over Australia.

To demonstrate how the locations of the modes and total occurrences vary with time of day, Figure 12 shows how the two modes of convection vary as a function of time of day with fits generated from Equation (1) at 2 hour intervals. Figure 12a-d show that total counts start to increase at around 1700 local as convection initiates and transitions from congestus to deep convection. At midnight, a decrease in the total number of counts is seen, suggesting that the loss of solar heating, an important factor during break conditions (May et al., 2012), is contributing to the decay of convection. The total counts and then show a second peak during the overnight hours in Figure 12ac with bimodal distributions. This secondary peak during the overnight hours, given the oceanic nature of the overnight convection as suggested by Figures 10e-h, and 11e-h are likely due to an increase in the number of MCSes at night during break conditions as indicated in Table 1. Figures 12bd show the overnight peak is less pronounced when the convective phase of the MJO is away from Australia. Since MCSs are larger in extent in the convectively active phase of the MJO as seen by (Virts and Houze, 2015) and more frequent as seen in Table 1, the reduced frequency of MCSs in the inactive phase of the MJO in Table 1 is consistent with these past observations. Rather, as suggested by the spatial analysis and Rauniyar and Walsh (2016), Hector and seabreeze convergence lines are more likely contributing to the distributions seen here. During monsoon conditions, Figure 12c shows that a majority of the counts occur when the

convective phase of the MJO is over Australia and during the daytime hours where the highest frequency of MCSes were identified. Similar conclusions about the peaks of convective activity seen in Figs. 12ab can be made in Figs. 12c. However, more unimodal distributions of ETH are seen in Figure 12d with little congestus present and few occurrences at night.

## 5    Conclusions

This study examined the macrophysical properties of convection in Darwin in differing phases of the MJO and Northern Australian Monsoon, including the echo top heights, convective areas, and number of mesoscale convective systems detected by CPOL during 17 wet seasons. The ETHs were generated using a methodology that uses velocity texture, differing from past studies that used reflectivity based thresholds such as May and Ballinger (2007); Kumar et al. (2013a, b). The use of velocity texture provides the potential for an automatic detection of the noise floor which increases the capability of including the lowest

reflectivties that still correspond to meteorological echoes and is also immune to radar miscalibration. It is demonstrated that such ETHs are correlated with CTHs retrieved by satellites (Ohkawara, 2004; Minnis et al., 2011) and that there is little sensitivity to ETH to whether reflectivity or velocity texture is used to retrieve the ETH, showing that retrievals of ETH are robust. In addition the ETHs of the lowest cloud layer detected by CPOL were 4 to 5 km lower than the VISST retrieved CTHs for CTHs > 10 km, showing that CPOL can detect and account for multi-layered clouds. These ETHs, along with convective

areas and the occurrences of MCSes, are then sorted by the Wheeler and Hendon (2004) MJO index and Drosdowsky (1996) monsoon/break classification. Some key conclusions can be made from this data:

1. Bimodal echo top heights were observed, and more common during break conditions, with a peak at around 3 to 4 km and another around 7 to 9 km, likely corresponding to cumulus congestus and deep convection. The break between these peaks corresponds with the presence of a stable layer at 5 km inhibiting the development of more intermediate convection.

2. Unimodal distributions were more common during an active monsoon, with an enhanced level of MCSes being observed during these periods as indicated by an increased average number of MCSes present. This is consistent with past studies suggesting that long lived MCSes are present during these conditions. The lower convective areas observed in active monsoon conditions, especially at night, indicate that these MCSes are generally weaker than those observed during break conditions. The MJO is a more important control of cloud top heights of deep convection than the phase of the monsoon for the 17 years

of data shown here, with lower echo top heights observed when the MJO is away from Australia. For the ETHs of congestus, we show that the monsoon is a greater control of the observed ETHs.

3. The convective cell areas were greater in break conditions than during monsoon conditions, but only when the active phase of the MJO was away from Australia. This difference was particularly present during the night. This shows that, when the active phase of the MJO is over Australia, the presence of the monsoon does not determine the size of the convective cells

and hence the strength of the updrafts. However, when the active phase of the MJO is away from Australia, we observe larger cells and hence stronger updrafts during the break phase of the monsoon.

4. The observed cloud top heights during the day in break and MJO-inactive conditions showed the presence of Hector and cumulus congestus. Meanwhile, at night, the distributions of echo top heights along with the analysis of the number of

MCSes observed showed that oceanic MCSes were more prevalent during the nighttime hours during break and MJO-inactive conditions.

5. The fewest occurrences of convection were observed during both an inactive MJO and an inactive monsoon. Given that there were more days observed during an inactive MJO and inactive monsoon than when the MJO and monsoon were active over Australia, this shows that convection is suppressed during these conditions.

The observed distributions of echo top heights and convective areas seen here create a suitable climatology for the validation of convective parameterizations in global climate models and are in the process of being used for the validation of the Department of Energy's Energy Exascale Earth System Model model. Future studies should focus on the improvement of the representation of the MJO and the monsoon in global climate models, as these results demonstrate the clear importance of both phenomena in determining the properties of convection observed in Northern Australia. It is also clear from the research presented here that the MJO is important for determining the properties of convection over Darwin and future studies looking at aspects of convection such as vertical velocities should consider the influence of the MJO as well as the monsoon.

*Code availability.* The code used to generate these plots is available at https://github.com/EVS-ATMOS/cmdv-rrm-anl/. The code used to generate velocity texture is included in the Python Atmospheric Radiation Measurement (ARM) Radar toolkit, available at http://github.com/ARM-DOE/pyart

*Data availability.* The data used to generate the cloud top height dataset is in the process of being submitted to the Atmospheric Radiation Measurement (ARM) Archive and will be available as a PI product upon publication of this manuscript.

*Competing interests.* There are no competing interests for this manuscript.

*Acknowledgements.* Argonne National Laboratory's work was supported by the U.S. Department of Energy, Office of Science, Office of Biological and Environmental Research, under Contract DE-AC02-06CH11357. This work has been supported by the Office of Biological and Environmental Research (OBER) of the U.S. Department of Energy (DOE) as part of the Climate Model Development and Validation activity. The development of the Python ARM radar toolkit, was funded by the ARM program part of the Office of Biological and Environmental Research (OBER) of the U.S. Department of Energy (DOE). The work Monash University and the Bureau of Meteorology was partly supported by the U.S. Department of Energy Atmospheric Systems Research Program through the grant DE-SC0014063. We gratefully acknowledge use of the Bebop cluster in the Laboratory Computing Resource Center at Argonne National Laboratory. The bulk of the code has been written using the open-source NumPy, Scipy, Matplotlib, Jupyter and Dask projects, and the authors are grateful to the authors of these projects. Special thanks are given to Brad Atkinson and Dennis Klau for the continual upkeep of the CPOL radar.

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

**Table 1.** The normalized frequency of MCSes identified using the criteria of (Rowe and Houze, 2014) in the radar domain per scan for the given MJO and monsoonal conditions.

| Time of day | Break | Monsoon | MJO away from Australia | MJO over Australia |
|---|---|---|---|---|
| Total | 0.17 | 0.30 | 0.14 | 0.24 |
| Day | 0.16 | 0.34 | 0.13 | 0.22 |
| Night | 0.18 | 0.26 | 0.14 | 0.24 |
| Time of day | Break and MJO away from Australia | Break and MJO over Australia | Monsoon and MJO away from Australia | Monsoon and MJO over Australia |
| Total | 0.13 | 0.21 | 0.20 | 0.33 |
| Day | 0.12 | 0.19 | 0.25 | 0.36 |
| Night | 0.13 | 0.23 | 0.13 | 0.30 |

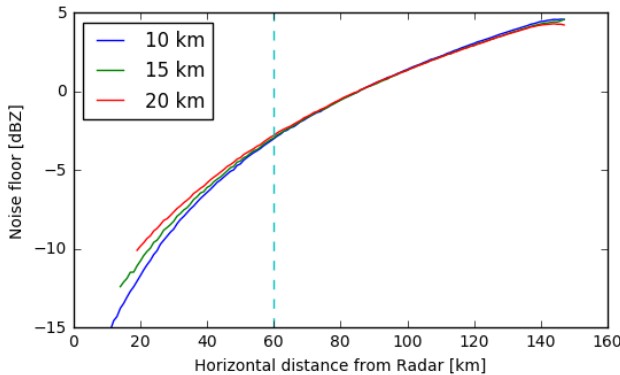

**Figure 1.** Minimum detectable signal of CPOL as a function of horizontal distance from CPOL for 3 vertical levels.

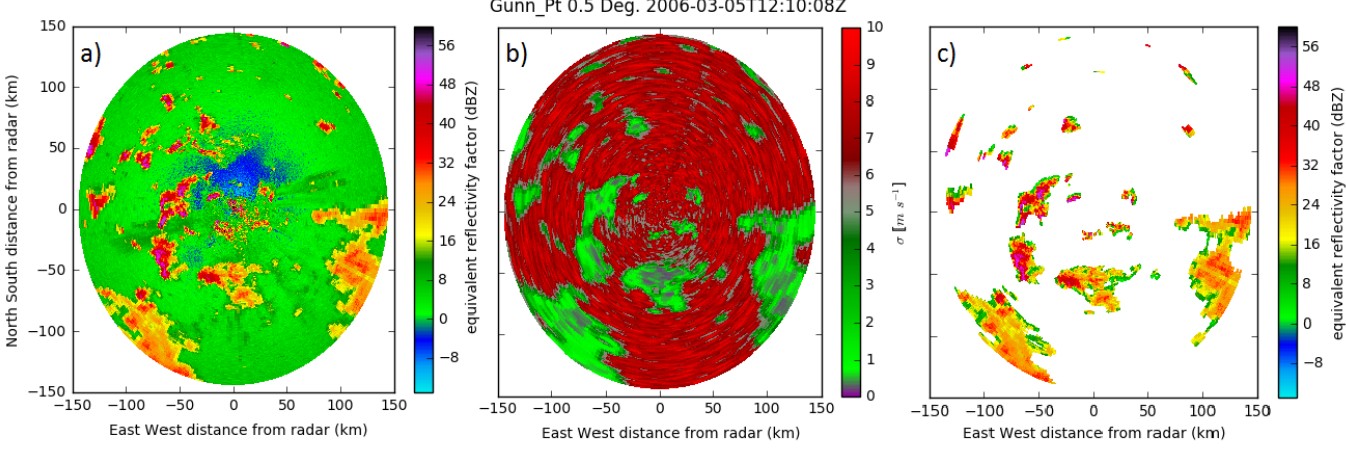

**Figure 2.** (a) Example Z (a) and (b) $\sigma$ field for a PPI scan from CPOL on 05 March 2006. (c) Z after masking gates with $\sigma > 3$

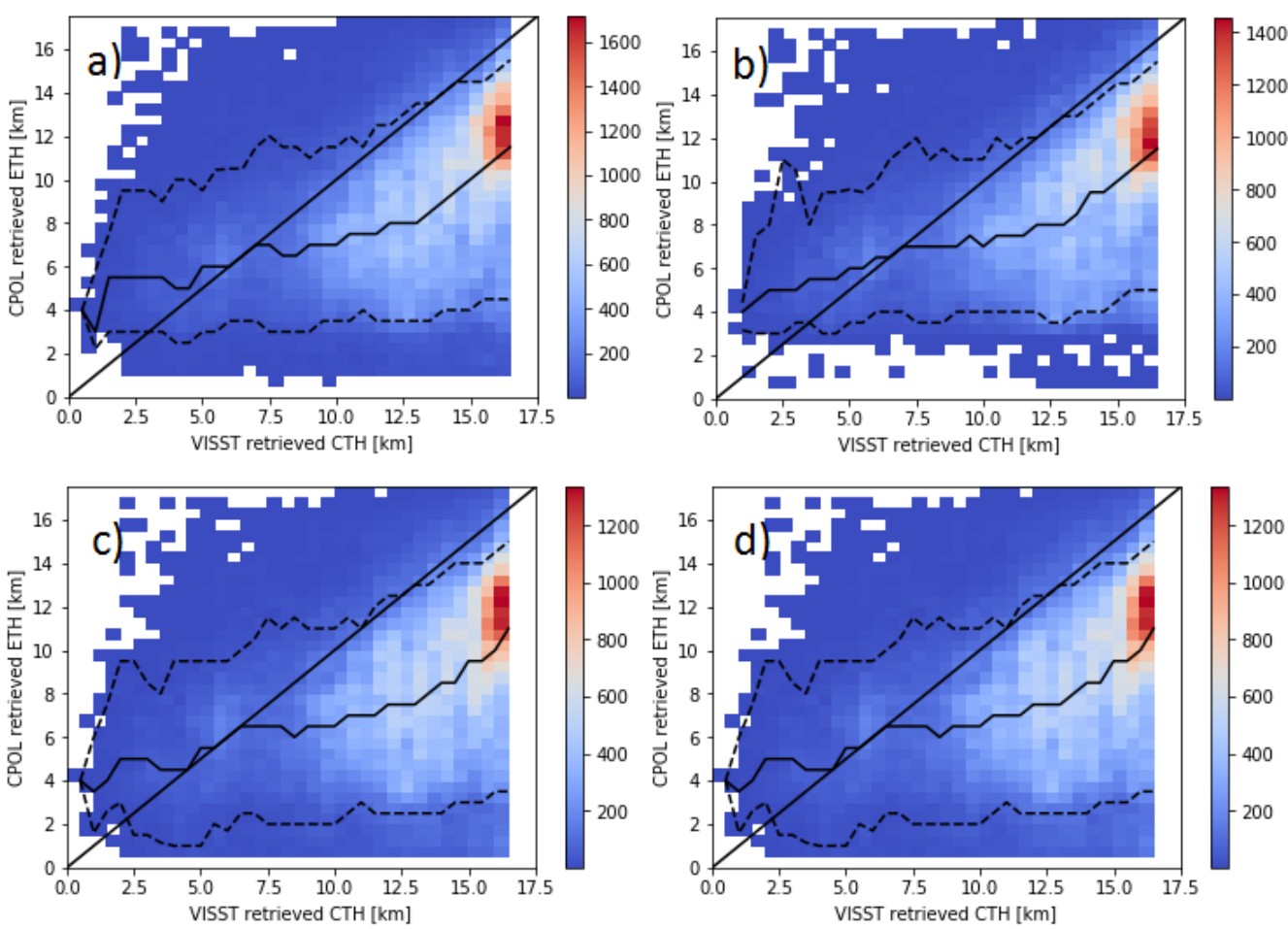

**Figure 3.** (a) ETH from CPOL retrieved using a Z threshold of 5 dBZ and (b) using a $\sigma$ threshold of $2\ m\ s^{-1}$. (c) using a $\sigma$ threshold of $3\ m\ s^{-1}$ and (d) $4\ m\ s^{-1}$ as a function of the VISST retrived CTH. The dashed lines represent the 5th and 95th percentiles of CPOL ETH, while the solid line represents the median of the CPOL ETH. The shading represents the number of occurrences.

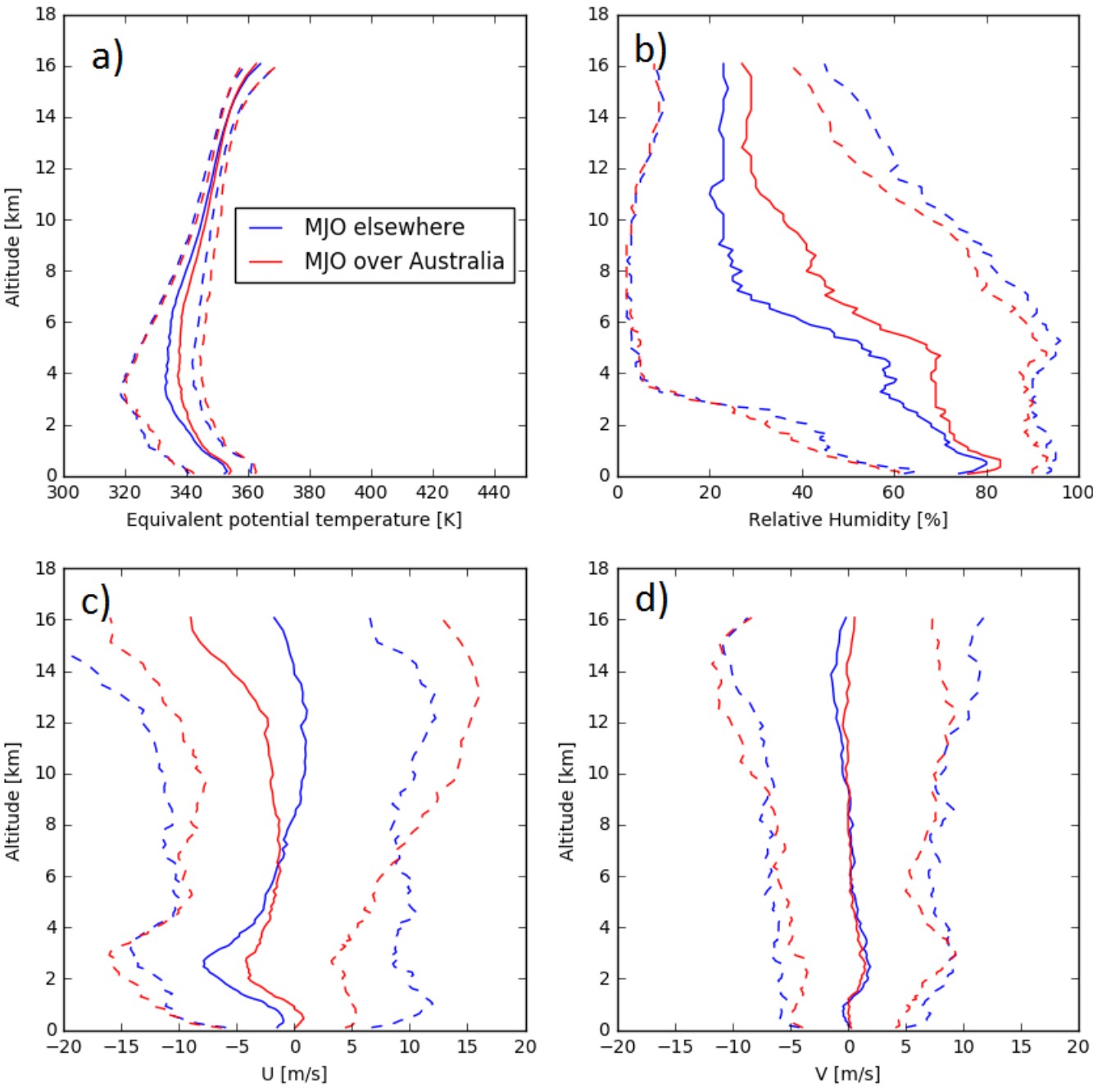

**Figure 4.** Mean (solid line) and 5th/95th percentiles (dashed lines) of (a) equivalent potential temperature, (b) relative humidity, (c) zonal wind and (d) meridional wind as a function of height from 949 rawinsonde observations during break conditions.

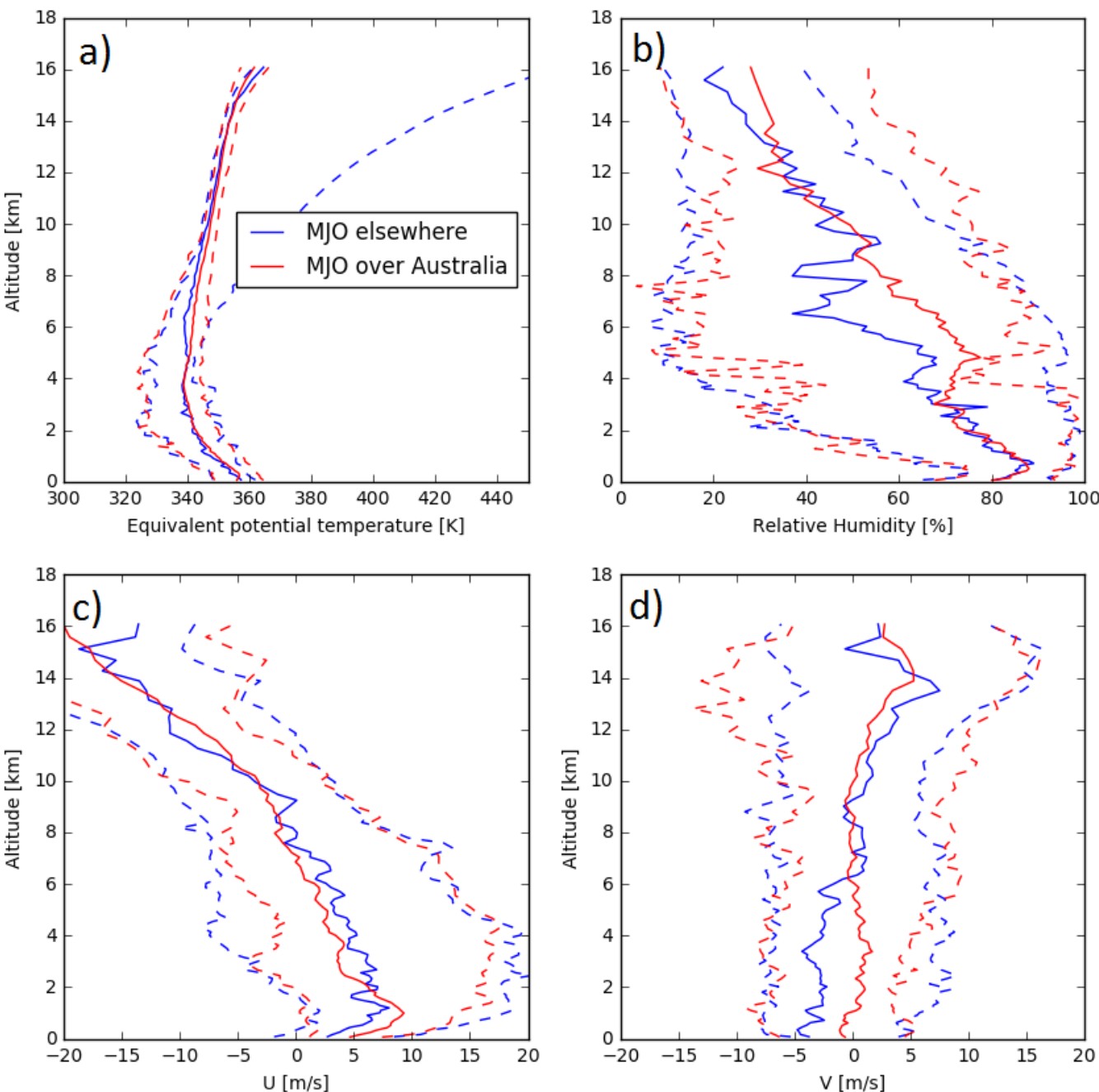

**Figure 5.** As Figure 4, but from 97 soundings taken during active monsoon conditions.

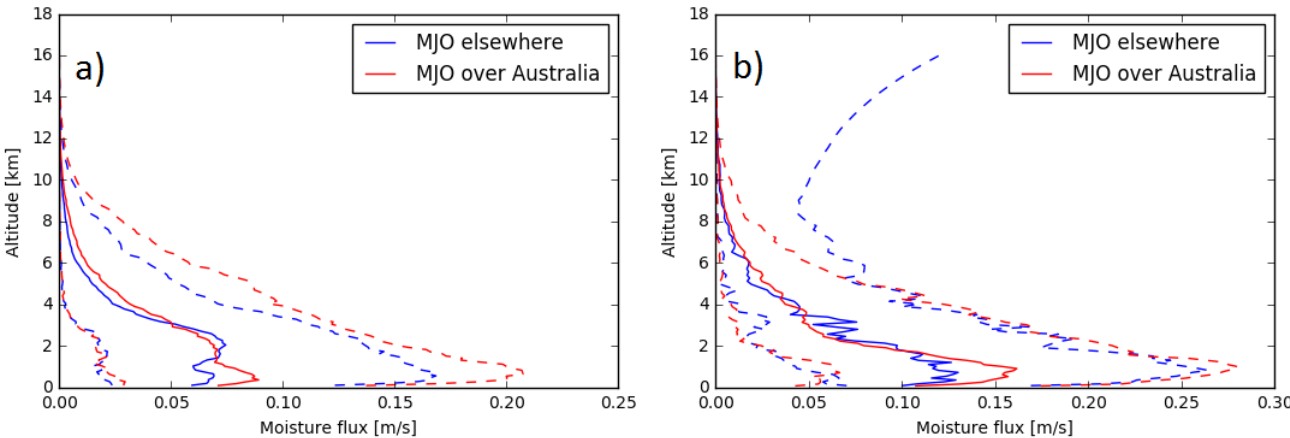

**Figure 6.** (a) As Figure 4, but showing moisture flux ($(u^2 + v^2)^{1/2}q$), where $q$ is specific humidity during break conditions. (b) as (a) but for active monsoon conditions.

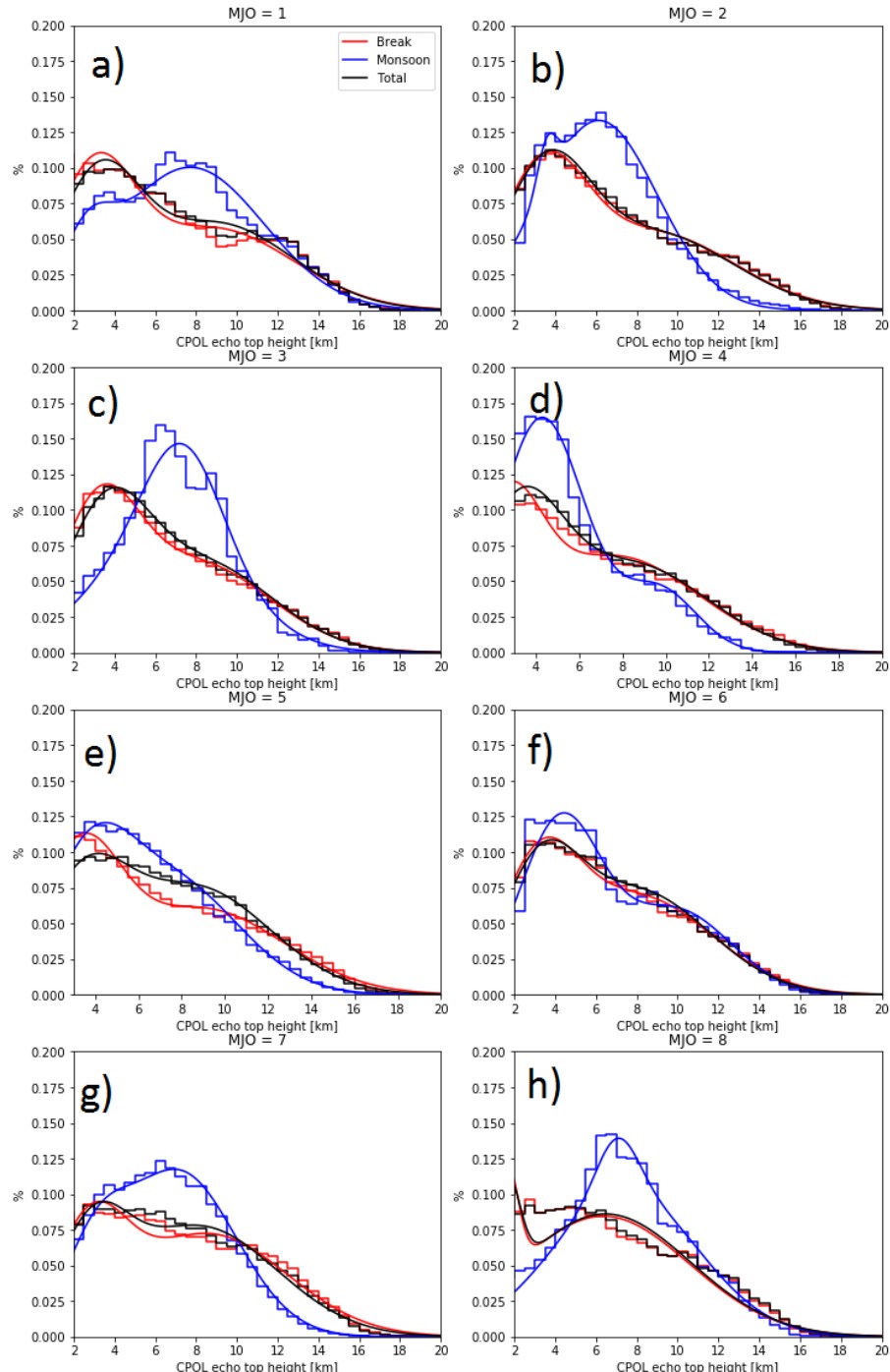

**Figure 7.** Normalized frequency distribution of ETHs in convective regions as a function of MJO index for break and monsoon conditions. Panels (a-h) each represent a different MJO index. Solid lines represent medians of modes derived from the fit of the bimodal Gaussian p.d.f. to the normalized frequency distribution.

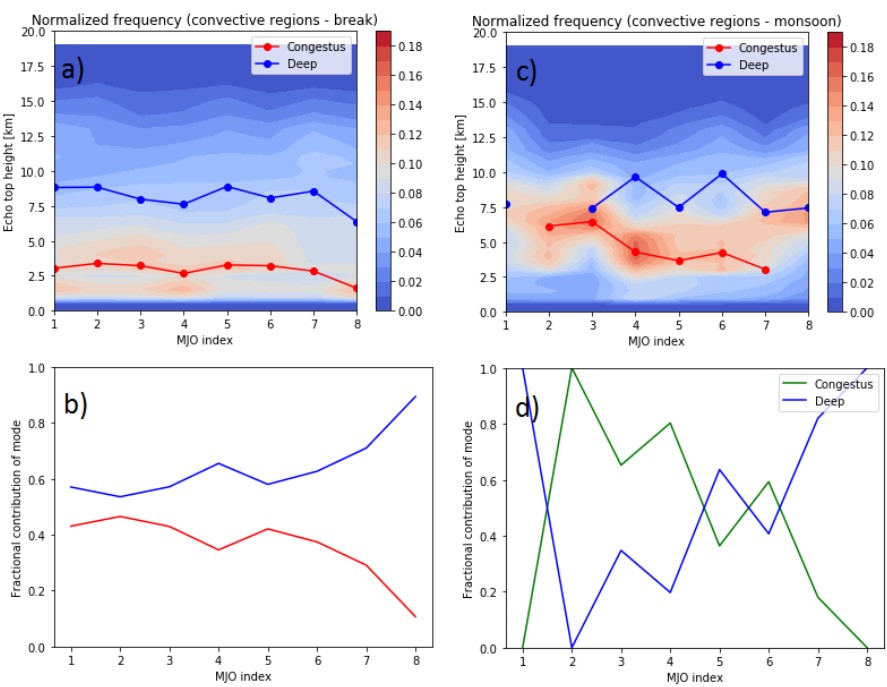

**Figure 8.** (a) Normalized frequency distribution of ETHs in convective regions for given MJO indicies in break conditions. The color shading represents the normalized frequency. The red line is the location of the congestus mode and the blue line is the location of the deep convective mode. (b) Fractional contribution of each mode ($A$) to normalized frequency distributions in (a). (c,d) as (a,b) but for monsoon conditions.

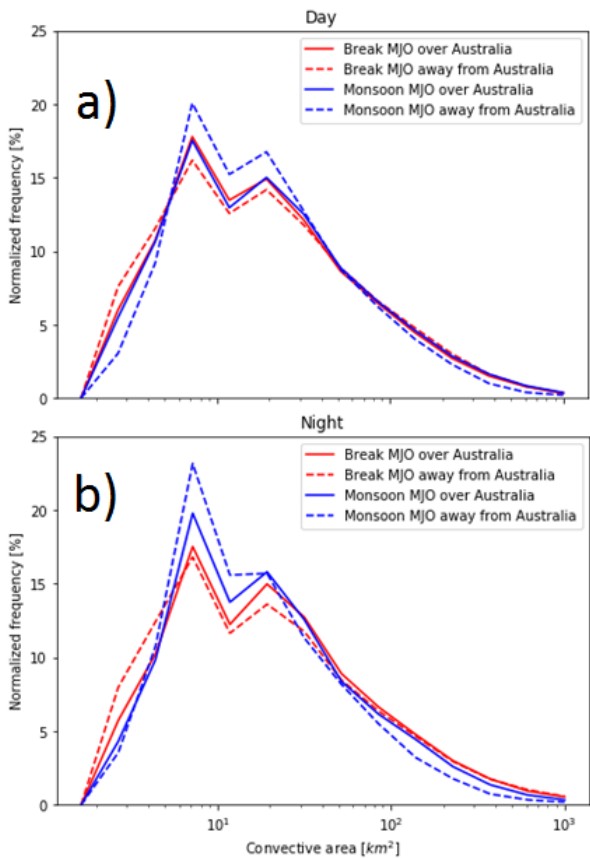

**Figure 9.** (a) Normalized frequency distribution of convective areas during the day (600 to 1900 local time). Each curve represents the large scale forcing specified in the legend. (b) as (a) but for night time.

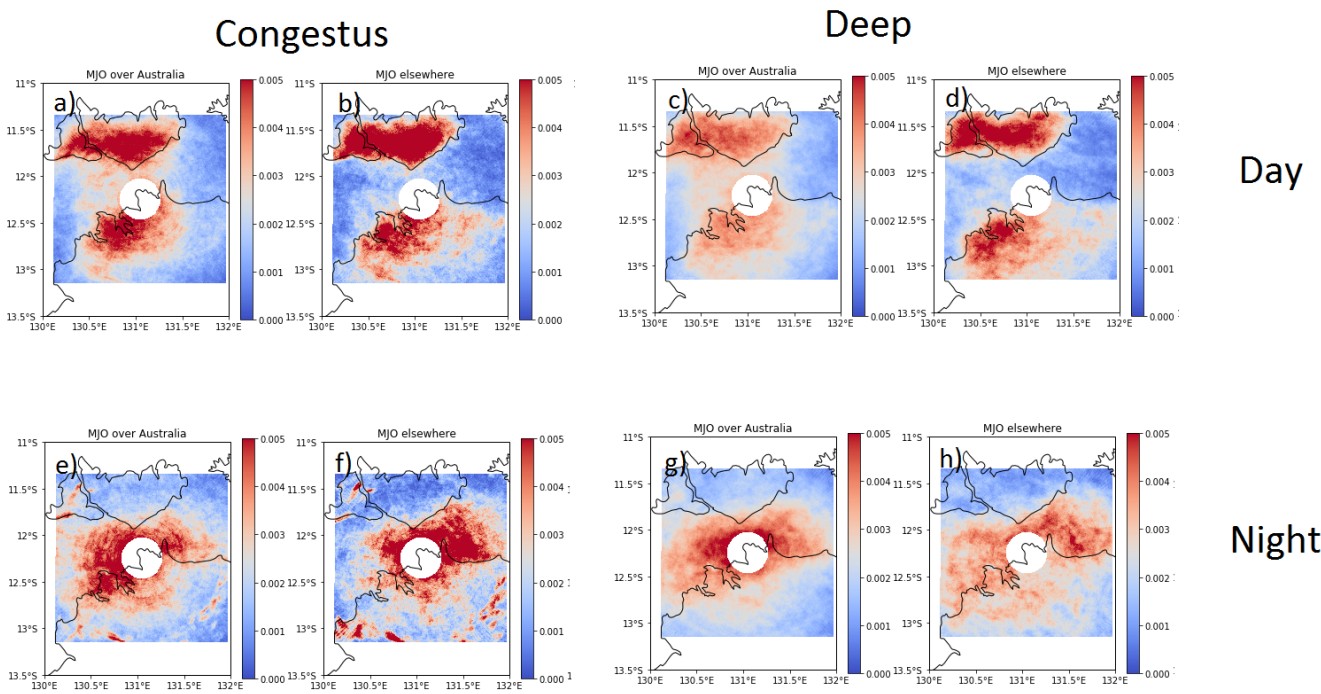

**Figure 10.** The normalized frequency of occurrence of ETHs < 7 km in MJO active (a) and (b) MJO inactive conditions during the day (600 to 1900 local). (c,d) as (a,b) but for ETHs > 7 km. (e,f) as (a,b), but at night 1900 to 600 local). (g,h) as (e,f), but for ETHs > 7 km. The histograms in Fig. 10 were taken during break conditions

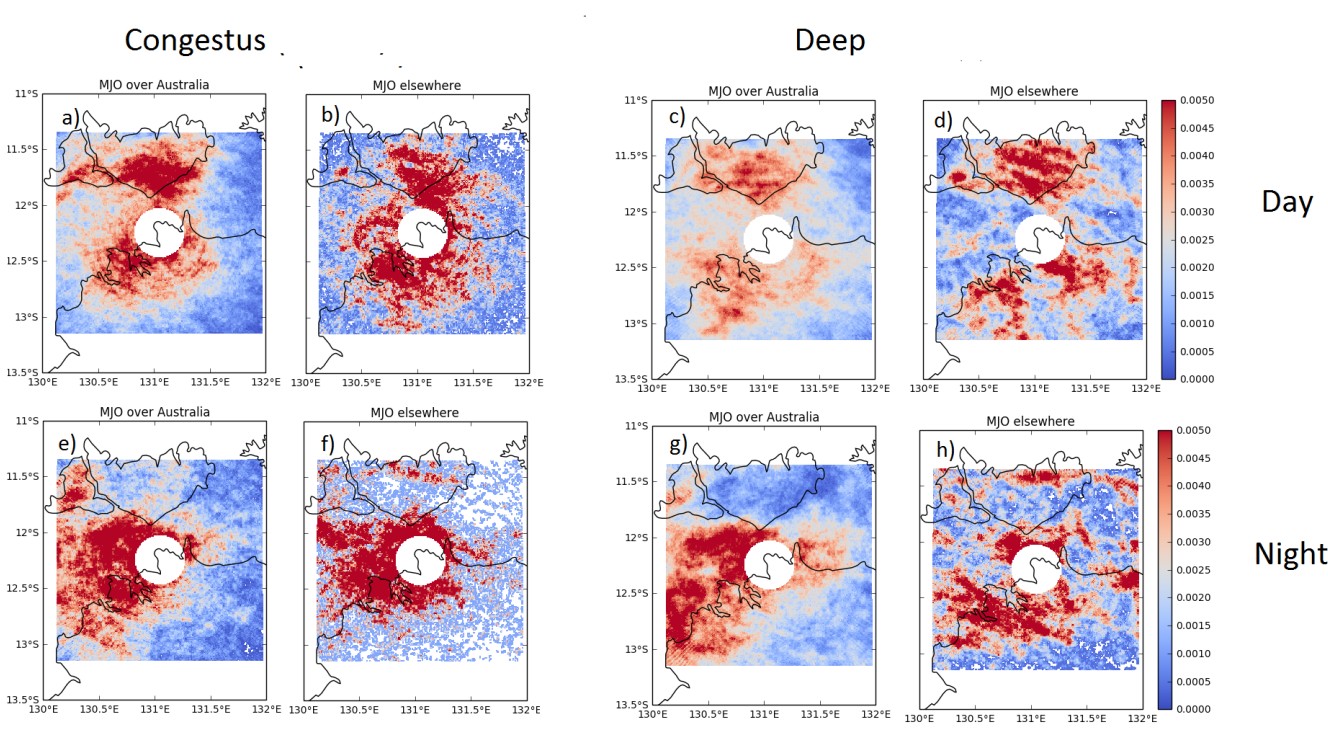

**Figure 11.** As Fig. 10, but for active monsoon conditions.

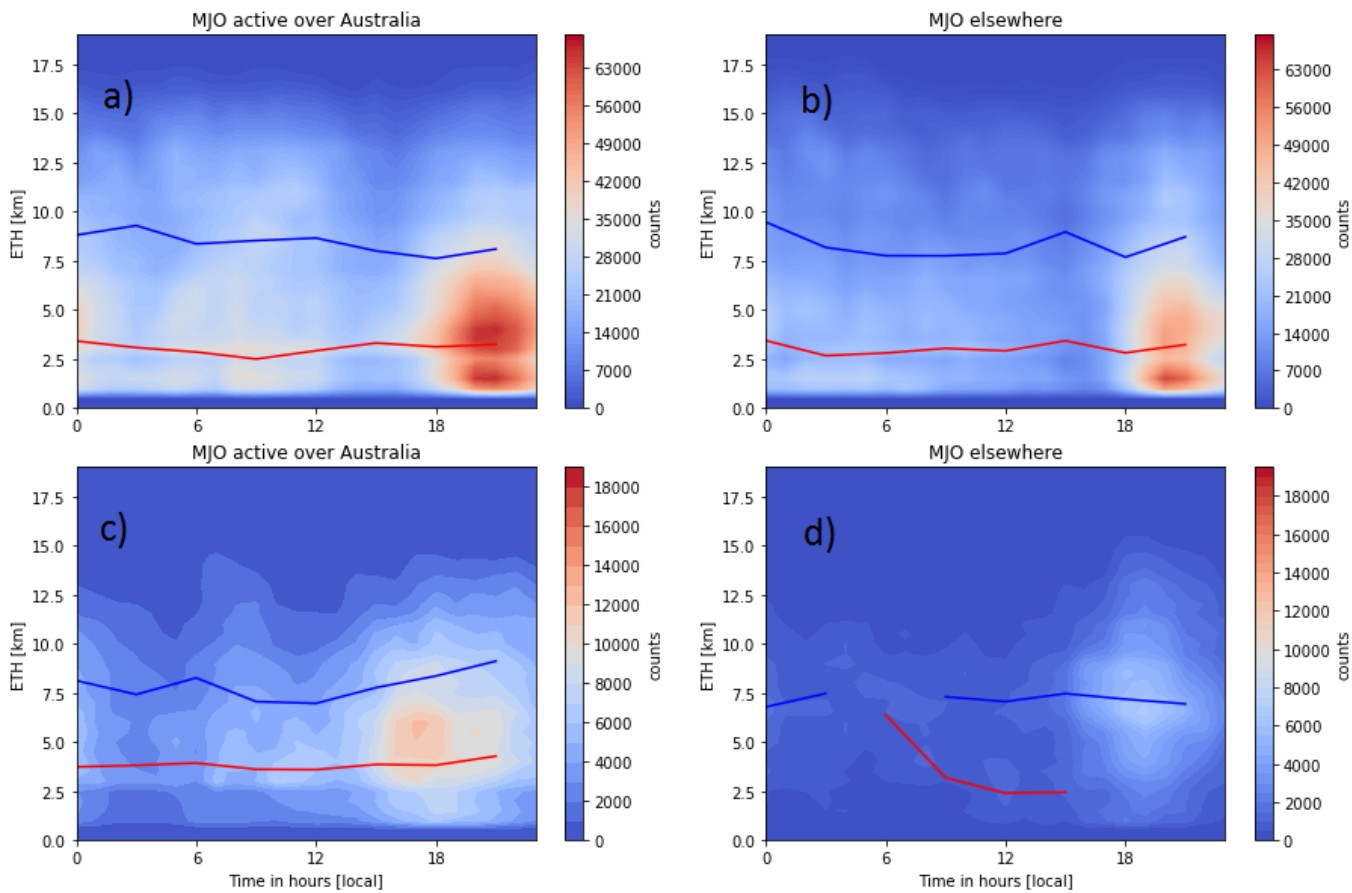

**Figure 12.** (a) Frequency distribution of occurrence of ETHs in convective regions as a function of time in break conditions when the MJO is active over Australia. Red line indicates peak of cumulus congestus mode, blue line indicates peak of deep mode of convection. (b) as (a) but for when the MJO is inactive over Australia (c,d) as (a,b) but during active monsoon conditions.