# Peer review of "A 17 year climatology of the macrophysical properties of convection in Darwin"

_Atmospheric Chemistry and Physics, 2018_

## Referee Comment (RC1) · Anonymous Referee #1 · 4 Aug 2018

**Review**: ACP-2018-408

Title: A 17 year climatology of convective cloud top heights in Darwin

**Recommendation: Major Revision.**

**Summary**

This study examines the variability of convective echo-top heights (ETH) observed by long-term CPOL radar in Darwin Australia as functions of large-scale conditions during active/suppressed MJO and monsoon/break periods. A new technique to estimate ETH is described and compared to a traditional reflectivity threshold based technique, and to a short period of geostationary satellite retrieved cloud-top heights. The study then continues to partition ETH distributions by combining MJO phases with monsoon

indices, and concludes that MJO has relatively stronger influence to convective ETH than monsoon.

In general, I think there are valuable results presented in this paper, particularly illustrating the MJO activity over Australia are relatively more important in regulating convective ETH compared to monsoon vs. break conditions, which previous studies have not investigated. However, there are many aspects of the study that need improvement to make it a significant contribution worthy of publication in ACP.

First, the "novel new technique" described in this study does not prove to be any better (or even different) than simpler existing method in calculating ETH, at least based on the short and in my opinion problematic comparison with a passive satellite cloud-top height retrieval dataset. Then why make a big deal about it? There is nothing wrong with using existing ETH technique, especially if you can't show that this new method works better than previous ones.

Second, I think the author need to connect the relative difference in large-scale conditions between active/suppressed MJO and monsoon/break to help explain why MJO has stronger modulation to convective ETH. Looking more at the difference in variability of the sounding profiles (rather than just the mean) between those conditions may be useful.

Third, the study misses an opportunity to examine how do these different large-scale regimes modulate an important aspect of convection in Darwin: the variability of convective cell sizes. Several recent studies have pointed out the importance of convective cell sizes to mass flux, a critical aspect to cumulus parameterizations. Also, analyses of spatial scales of convection would provide more concrete conclusions on changes of MCS activities with large-scale regimes, which the paper makes many reference but did not show supporting evidence.

I provided more detail comments below on various places in the paper that need improvement. Because I do see the value of the long-term tropical radar observations, I

recommend major revision of the paper before it can be accepted for publication.

**Major Comments**

1. Page 2 line 20, please be more specific on what aspects of convective parameterizations are "poor". Poor in doing what?

2. Section 3.1, what kind of quality control procedures were applied to the raw radial radar data? How do you handle ground clutter, AP, and noise that are particularly prevalent in CPOL data at lower level, which could affect your ETH estimates?

3. Page 4 line 18, why choose a "box" when the radar scans are circular? The diagonal corners of the box are 140 km away from CPOL, which is much further than 100 km radius where sampling and resolution are better.

4. Page 5 paragraph 2 and 3, I do not understand how is ETH estimated in CPOL using the Doppler velocity standard deviation ($\sigma$) technique. Figure 2 shows example of using $\sigma > 3$ to remove non-precipitating radar echoes, but then how is ETH determined from the remaining echoes? How is it different from simply just using Z threshold > 5 dBZ?

5. Please also clarify how is ETH estimated using the Z > 5 dBZ threshold method. Do you go from surface upwards and find the first height level where Z drops below 5 dBZ? Or do you go from top downwards to find first level where Z exceeds 5 dBZ (i.e. max height of 5 dBZ in a column)? They could give very different results because the second approach would get cirrus/anvil clouds that are above precipitating convective cloud-tops (particularly for existence of multi-layer clouds).

6. Figure 3, the authors did not provide enough detail about how the CPOL data and MTSAT data are matched for the comparison. Given the two datasets have different spatial resolution, do you match them by interpolating one to another? Do you only compare grid points that both CPOL and MTSAT identified as echo/cloud?

7. There is also the issue of daytime vs. nighttime retrieval differences in the MTSAT data. As I mentioned in minor comment 7, there are two satellite retrieval algorithms separately for daytime and nighttime. My previous experience working with these datasets are they do not necessarily provide consistent cloud-top height retrievals when switching from one to another. Cloud-top heights from the same cloud systems can differ as much as several kilometers between the two estimates, and during twilight hours (+/- 1-2 hour) when the solar zenith angle is high, the retrievals uncertainties are very large. Did you 1) compare daytime vs. nighttime separately? 2) exclude twilight hours?

8. Why do you choose such a short period of only two months during the peak monsoon TWP-ICE period for a comparison? Particularly when most of the precipitating clouds are deeper than 7-8 km. MTSAT retrievals at this location are available for multiple years in the ARM data archive. Further, wouldn't a more direct comparison be made between CloudSat measured cloud-top heights as it is active remote sensing? For such a long CPOL record and decent spatial coverage, there should be plenty of samples to compare. One of the coauthors of the study (Alain Protat) has made much harder comparisons between CloudSat and ARM cloud radars before (Protat et al. 2014), what stop you from doing that? I understand it takes some effort to do that, but if the paper wants to claim that this new method of estimate ETH from CPOL is closer to actual cloud-top height, then a more stringent evaluation is needed than what is presented here.

9. Page 5 lines 26-29, if the new technique using $\sigma$ to calculate ETH (which I do not understand, see major comment 4) gives a similar result with a much simpler $Z$ threshold approach, what is so unique about this new approach then if it does not perform better than existing ones? Why make a big deal about the novelty?

10. Page 6 line 30, I cannot tell if there is a significant difference in dew point temperature between monsoon break and active period from Fig. 3-4 as they look very similar. You should also compare specific and relative humidity profiles. Mid-level humidity in the tropics is particularly important for supporting deep convection (e.g. Hagos et al. 2014). Showing the difference between the profiles may be useful. Also, given the

small difference in the mean thermodynamic profiles between MJO phases, showing a difference in the mean and the variability is useful as well. Please also include the number of soundings that go into the composite.

11. Figure 6, can you comment on why in this study the "overshooting" mode found in Kumar et al. (2013) is not visible in the much longer dataset? Their study showed that the overshooting mode correspond to intense low-level reflectivity (and inferred larger and more numerous raindrop particles), which tend to occur more during the monsoon break period. Does the 14 km peak in break period (Fig. 6 MJO=1,2) correspond to that mode?

12. The congestus mode in Kumar et al. (2013) is defined as ETH < 6.5 km, where in this study it is 8 km. That is not a small difference. 8 km is also significantly higher than the 0C ( 4.5-5.0 km) level where above which freezing and additional latent heating acceleration of vertical motion can occur. Can you comment on why you choose a larger ETH value for congestus?

13. Page 8 line 16-18, I thought A is the contribution of mode 1 (congestus), when A increase to 0.9 during active phases of the MJO under monsoon conditions, doesn't that mean most of the convection are congestus, as opposed to deep convection stated in this sentence? That is contrary to the statement of mostly widespread MCSs during active MJO and monsoon. I think the issue here is using only echo-top height to indicate congestus vs. deep convection is too simplistic. The wide spread MCSs are likely associated with much larger but not as deep convective cells compare to more isolated but deep convection during break period. One very important indicator for organized convection, i.e., the size of convective cells is ignored in this study. Larger convective cells and larger updrafts (i.e. in MCSs) carry a majority of the mass flux as reported in both observational analyses (Kumar et al. 2015, Masunaga and Luo 2016) and high-resolution model results (Hagos et al. 2018). The size of convective cells is just as important (if not more) as the depth of the convective cells. It can be easily quantified with the CPOL data and I think would be a useful quantity to investigate as functions of

large-scale regimes along with the ETH analyses.

14. Page 8 line 29-30, I think this is an interesting and important finding. It would be useful to discuss what aspects of the large-scale conditions can help explain larger difference under active MJO (i.e. going back to Fig. 4-5, see major comment 10 about quantifying their relative environment profile difference between active/suppressed MJO, monsoon/break).

15. Figure 9, other than the more obvious enhanced frequency of deep convective during daytime, the difference for the rest of the panels between MJO phases are difficult to see. Perhaps adding a difference panel would help.

16. Page 9 line 26-29, why do you have to guess that the enhanced nighttime peak over the ocean during monsoon period is due to MJO? It is relatively easy to identify MCSs in CPOL data (e.g., Rowe et al. 2014 used a simple criteria of precipitation feature major axis length > 100 km to identify MCS in ground-based radar observations), why not actually quantify MCS frequency changes to better support your claim?

17. Figures 10-11, given the strong diurnal cycle between land vs. oceanic area shown in Figures 8-9, did you separate land area and ocean area when calculating their ETH occurrence? I also suggest plotting Figures 10-11 in local time to make it easier for readers.

18. Page 11 line 13-14, which figure shows a peak of ETH around 5-6 km during break conditions? Figure 6 when MJO is away from Australia (phases 1-2) generally show a peak between 6-8 km, but drops to 4 km in phases 3-4.

**Minor Comments**

1. Page 1 line 2, technically validation of convective processes in GCMs do not "require" such statistics, perhaps it's better to say "could benefit from" such statistics.

2. Page 1 line 5, why does it have to be for a specific model? These observations can be useful for any large-scale model validations.

3. Page 1 line 22, Jensen et al. (1994) stated the 100 W/m2 is solar forcing, not net radiative forcing, please clarify that in the statement.

4. Page 2 line 19, "... of of an intense ...".

5. Page 2 line 27, "... for one wet season in and found ..." in what?

6. Page 4 line 1, spell out the acronym "ACRF".

7. Page 4 line 4-6, I believe the MTSAT data, at least the inferred channel spatial resolution should be 5 km, not 1 km. Also, the VISST technique uses all available geo-stationary satellite channels, including visible, water vapor, near IR, and IR channels to retrieve cloud properties during day time. For nighttime, a different technique SIST is used for the lack of visible channel.

8. Page 5 line 14, you mentioned "normalized frequency distribution", what is the unit of the shading in Figure 3? The large numbers appears to be just a count, if so it is not normalized frequency.

9. Page 5 line 31, do not use "cloud top height" and ETH interchangeably. You have not established in this study that CPOL ETH is equivalent to cloud top height.

10. Figs. 3-4, why is the vertical scale different between the two figures?

11. Page 7 line 12, spell out p.d.f. in section headings.

12. Page 8 line 8, "In 7, ..." do you mean in "Eq. (1)"?

**References**

Protat, A., S.A. Young, S.A. McFarlane, T. L'Ecuyer, G.G. Mace, J.M. Comstock, C.N. Long, E. Berry, and J. Delanoë, 2014: Reconciling Ground-Based and Space-Based Estimates of the Frequency of Occurrence and Radiative Effect of Clouds around Darwin, Australia. J. Appl. Meteor. Climatol., 53, 456–478, https://doi.org/10.1175/JAMC-D-13-072.1

Kumar, V. V., C. Jakob, A. Protat, C. R. Williams, and P. T. May (2015), Mass‐flux characteristics of tropical cumulus clouds from wind profiler observations at Darwin, Australia, J. Atmos. Sci., 72, 1837–1855, doi:10.1175/JAS-D-14-0259.1.

Masunaga, H., and Z. J. Luo (2016), Convective and large‐scale mass flux profiles over tropical oceans determined from synergistic analysis of a suite of satellite observations, J. Geophys. Res. Atmos., 121, 7958–7974, doi: 10.1002/2016JD024753.

Hagos, S., Z. Feng, K. Landu, and C. N. Long (2014), Advection, moistening, and shallow-to-deep convection transitions during the initiation and propagation of Madden-Julian Oscillation, J. Adv. Model. Earth Syst., 06, doi:10.1002/2014MS000335.

Hagos, S., Feng, Z., Plant, R. S., Houze, R. A., Xiao, H. (2018). A stochastic framework for modeling the population dynamics of convective clouds. Journal of Advances in Modeling Earth Systems, 10. https://doi.org/10.1002/2017MS001214.

Rowe, A. K., and R. A. Houze Jr. (2014), Microphysical characteristics of MJO convection over the Indian Ocean during DYNAMO, J. Geophys. Res. Atmos., 119, 2543–2554, doi: 10.1002/2013JD020799.
* * *

---

## Referee Comment (RC2) · Anonymous Referee #2 · 11 Aug 2018

General Comments This study aims to define characteristics of monsoon phase (break vs active) within context of MJO phase over Australia for northern AU region. 17 years of radar data are used to increase sample size and develop statically significant results. Overall I think this is an interesting study that shows the impact of MJO phase on cloud top heights for active and inactive periods drawing on precipitation and thermodynamic characteristics to explain the results.

My major criticism is that the description is confusing and hard to follow in parts of the statistical analysis (Sec. 4.1) and diurnal cycle (Sec. 4.2) sections as described in the specific comments below. The confusion is due to 1) combination of too many variables to consider when trying to correlate interpretations stated in the text to results shown in selected figures: active vs inactive monsoon, MJO phase, day vs night, ocean vs land;

[Figure]

and 2) the text is not always explicit in terms of which panel of a figure is being used to advance the argument. In consequence, I as the reader, have sometimes come to a different conclusion when interpreting the figures in question compared to the authors. I have also noted these instances below.

A more modest critique concerns the spectrum width thresholding technique used to discriminate echo top height as opposed to a minimum reflectivity threshold. On the bright side, the method appears to work reasonably well. However, the end results is that there does not seem to be any real difference in the results when compared to simply applying a minimum reflectivity threshold (which is the traditional approach) so I am left scratching my head when trying to understand the real advantage of the methodology.

Specific Comments 1. P. 4, line 12; please define gate spacing and resolution 2. P. 5, line 7: What is the spatial resolution of the satellite data? If it's less than radar resolution it's not clear what a relative comparison tells us regarding the performance of the radar-based ETH algorithm. 3. P. 5, line 15: similar to previous comment - to understand the differences in cpol vs satellite – what is satellite brightness temp keying off of – what depth of cloud is considered? 4. P. 5, line 17: cc of 0.49 is not very good 5. P. 5, line 20: this statement assumes the satellite is capturing the variability. . . 6. Fig. 3 please state in the caption what the color shading represents 7. P 6. Lines 25-30: In references to Figs 4-5, seems like the big differences are between monsoon phase instead of MJO phase? 8. P. 7, line 7 – There are several other older references that show this behavior: Cifelli and Rutledge 1994 (JAS); 1998 (QJRMS) 9. P. 7, line 27-28: some hint of trade wind layer in MJO=3 for break (Fig. 6)? 10. P. 8, line 8: This is a minor point but it should be noted that the heights of the different modes that are stated here are approximate. For example, in Fig. 7c the height of mode 2 does not appear to actually reach 15 km. . . 11. P. 8, line 12-18: there is some confusion looking at Fig 7. My read of the red line (A=congestus) in MJO phases 4-7 is $\sim$0.05 − 0.6 for break (Fig 7b) – not 0.8-0.5 as described - and $\sim$0.1-0.4 for monsoon (Fig. 7d) –

not > 0.9 as described. Also, the statement on line 14 about unimodality is confusing: Fig 7b,d show that there is a significant contribution from the congestus mode in break conditions while the MJO is over AU (Fig. 7b, MJO phases 6,7). Similar in monsoon conditions for MJO phase 6 – see Fig 7d. I think the confusion noted above could be avoided by stating more clearly which features in specific figure panels are being referred to. 12. Fig. 8 – please state in the caption and the figure that this is for break conditions 13. P. 9 -please call out panels explicitly in reference to Figs. 8-9 14. The discussion jumps to Fig 10 before discussing Fig. 9 15. P. 9, line 24: which panel of Fig 10? My read of comparing Fig 10 a and Fig 10b is that during the day there is a higher frequency of deep convection when the MJO is over AU (assume that includes Tiwi islands as well) compared to when MJO is elsewhere. 16. P. 9, 25-26: I don't understand the point about what is being extended in this study vs previous work.. 17. P. 10 lines 11-12 – where do the number of days come from?

---

## Author Comment (AC1) · 21 Sep 2018

Review
: ACP-2018-408
Title: A 17 year climatology of convective cloud top heights in Darwin
Recommendation: Major Revision.

Summary
This study examines the variability of convective echo-top heights (ETH) observed by long-term CPOL radar in Darwin Australia as functions of large-scale conditions during active/suppressed MJO and monsoon/break periods. A new technique to estimate ETH is described and compared to a traditional reflectivity threshold based technique, and to a short period of geostationary satellite retrieved cloud-top heights. The study then continues to partition ETH distributions by combining MJO phases with monsoon indices, and concludes that MJO has relatively stronger influence to convective ETH than monsoon.

In general, I think there are valuable results presented in this paper, particularly illustrating the MJO activity over Australia are relatively more important in regulating convective ETH compared to monsoon vs. break conditions, which previous studies have not investigated. However, there are many aspects of the study that need improvement to make it a significant contribution worthy of publication in ACP.

First, the "novel new technique" described in this study does not prove to be any better (or even different) than simpler existing method in calculating ETH, at least based on the short and in my opinion problematic comparison with a passive satellite cloud-top height retrieval dataset. Then why make a big deal about it? There is nothing wrong with using existing ETH technique, especially if you can't show that this new method works better than previous ones.

**We would like to thank the reviewer for their insightful comments on this manuscript. In a phase randomized radar significant returns have a "Smooth" radial velocity while regions of no return or multi-path have radial velocity that varies randomly from -nyquist to +nyquist. This allows texture of radial velocity to provide a very "clean" determination of a significant return. Most importantly it provides a consistent definition of echo top height rather than relying on an arbitrary threshold. Texture allows the echo top height to match the height of minimum detectable signal.**
**We have rephrased the "novel new technique" wording in the abstract to simply state that we are assessing the applicability of such a methodology. This therefore reduces the scope of this section from attempting to state which methodology better estimates the cloud top height to simply a comparison that assesses the sensitivity of our results to the ETH retrieval technique used. We still compare the retrieved ETHs against cloud top heights estimated from satellites in order to assess whether or not the retrieval can capture the statistical variability in ETHs without necessarily knowing the uncertainty. Given a dataset as large as this, this approach is feasible as this tells us that we can observe the relative interseasonal variability in ETHs. Furthermore, we feel that it is**

**important to show the sensitivity of the ETHs to the retrieval technique used, which few studies do, even if a null result is shown.**

**Given the revision in scope of Section 3, the new wording in the abstract rephases the testing to demonstrate that there is little sensitivity to the ETH with retrieval technique and that our retrieval captures the relative seasonal variability in cloud top height:**

*"Retrieved ETHs are correlated with those from MTSAT retrieved cloud top heights, showing that the ETHs capture the relative variability in cloud top heights over seasonal scales. "*

**The wording in Section 3 regarding the methodology comparison already stated that both methods were roughly equivalent, so no changes were made there. However, since both methodologies give similar echo top heights as shown in the comparison in Section 3, we still elect to use the velocity texture methodology since it provides comparable results to using reflectivity.**

Second, I think the author need to connect the relative difference in large-scale conditions between active/suppressed MJO and monsoon/break to help explain why MJO has stronger modulation to convective ETH. Looking more at the difference in variability of the sounding profiles (rather than just the mean) between those conditions may be useful.

**We have added 5th and 95th percentiles to Figures 4 and 5 to represent the variability seen in each of the large scale forcing regimes and also now add specific humidity. We also now better connect the differences between the large scale forcings in the discussion of Figures 4 and 5 by examining the differences in the distributions of winds and specific humidities, with two paragraphs in Section 3.3 instead of one demonstrating the relative differences between the regimes. In it is shown in this section now that:**
1. **There is a greater variability in the Surface to 500 hPa winds during an inactive MJO, which contributes to fewer cases where stronger flow of moisture from the west are occurring.**
2. **There is reduced mid-level moisture (4 to 8 km) during inactive MJO/break conditions, as well as wider variability of such moisture as indicated by the specific humidity profiles. This suggests that the large scale environment during an active MJO is more favorable for the transition of congestus to deep convection.**

Third, the study misses an opportunity to examine how do these different large-scale regimes modulate an important aspect of convection in Darwin: the variability of convective cell sizes. Several recent studies have pointed out the importance of convective cell sizes to mass flux, a critical aspect to cumulus parameterizations. Also, analyses of spatial scales of convection would provide more concrete conclusions on changes

of MCS activities with large-scale regimes, which the paper makes many reference but did not show supporting evidence.

**We have added an analysis of both how the cell sizes vary and now objectively quantify the number of MCSes using Rowe and Houze (2014)'s methodology to justify our claims about MCS coverage. In addition we have added material to the introduction now introducing why cell size is important and what past studies have concluded about how the cell sizes in Darwin vary in differing large scale forcings in the introduction. We thank the reviewer for this suggestion!**

I provided more detail comments below on various places in the paper that need improvement. Because I do see the value of the long-term tropical radar observations, I recommend major revision of the paper before it can be accepted for publication.

Major Comments
1. Page 2 line 20, please be more specific on what aspects of convective parameterizations are "poor". Poor in doing what?

**We have replaced this sentence to be more specific by what we mean by "convective parameterizations are poor:"**

*Also, convective parameterizations in GCMs do not account for mesoscale organization resulting in insufficient sensitivity to upper tropospheric humidity (Del Genio, 2012).*

2. Section 3.1, what kind of quality control procedures were applied to the raw radial radar data? How do you handle ground clutter, AP, and noise that are particularly prevalent in CPOL data at lower level, which could affect your ETH estimates?

**The great advantage of the use of velocity texture compared to using reflectivity is that the noise floor can be detected automatically since we expect the velocity field in regions of noise and second trip echoes to be random, resulting in higher velocity textures. We are also doing the following steps:**

- **Excluding gates with differential reflectivity < -3 and > 7 dB.**
- **Excluding gates with < 0.45 cross-correlation coefficient.**
- **Excluding gates with differential phase texture > 20**
- **Excluding gates with reflectivity less than -20 dBZ and greater than 80 dBZ.**

**We have added these steps into a bulleted list into Section 3.1**

3. Page 4 line 18, why choose a "box" when the radar scans are circular? The diagonal

corners of the box are 140 km away from CPOL, which is much further than 100 km radius where sampling and resolution are better.

**We chose a "box" since we interpolated the radial data onto a Cartesian grid for easier spatial analysis and using boxes is easier with data in Cartesian coordinates. While creating the Cartesian grid, we used Barnes (1964)'s weighting function with a radius of influence that increases with distance from the radar in order to account for the decreased sampling and resolution as a function of distance from the radar. We conducted visual analyses of such grids and had determined that using a 100 km by 100 km box gave reasonable coverage, even at ranges of 140 km.**

4. Page 5 paragraph 2 and 3, I do not understand how is ETH estimated in CPOL using the Doppler velocity standard deviation ($\sigma$) technique. Figure 2 shows example of using $\Sigma > 3$ to remove non-precipitating radar echoes, but then how is ETH determined from the remaining echoes? How is it different from simply just using Z threshold > 5 dBZ?

**Previously, the ETH was determined by looking at the gate above the highest valid gate after the $\sigma$ technique was applied. The ETH is now determined by looking for the first echo in the column that is masked using the $\sigma$ technique. We have added some clarifying wording in this paragraph to explain how we retrieve the ETHs from the masked data: "***The ETHs are then determined by looking at the lowest gate in the column that is masked. We use the lowest gate in the column in order to ensure that we are capturing the ETHs of the precipitating convection, and not that of detrained anvils and cirrus that can lie above the precipitating convection.***"**

**This is different from using Z > 5 dBZ as we are not using reflectivity at all.**

5. Please also clarify how is ETH estimated using the Z > 5 dBZ threshold method. Do you go from surface upwards and find the first height level where Z drops below 5 dBZ? Or do you go from top downwards to find first level where Z exceeds 5 dBZ (i.e. max height of 5 dBZ in a column)? They could give very different results because the second approach would get cirrus/anvil clouds that are above precipitating convective cloud-tops (particularly for existence of multi-layer clouds).

**We had originally used the topmost point in the column where Z > 5 dBZ, or the second approach. However, in order to account for cases of multi-layer clouds, we have switched to the first approach as it is more representative of the precipitating convection. We now also make it more clear how we are deriving the ETH from Z in Section 3.**

6. Figure 3, the authors did not provide enough detail about how the CPOL data and MTSAT data are matched for the comparison. Given the two datasets have different spatial resolution, do you match them by interpolating one to another? Do you only compare grid points that both CPOL and MTSAT identified as echo/cloud?

**We interpolated the MTSAT data to CPOL's grid and now say so in Section 3.2. In addition we also state that we only compare points that were identified as in cloud by the VISST product and as convective by the Steiner et al. (1995) algorithm in CPOL.**

*"Since the two datasets are at differing resolution, the MTSAT data are interpolated onto the same grid as the CPOL data for the comparison. Furthermore, to ensure that we are comparing points that are in precipitating convection we both only include points from MTSAT where the VISST product identified cloud and where the convective classification algorithm, detailed in Section 3.4, classified the grid points as precipitating convection."*

7.   There is also the issue of daytime vs. nighttime retrieval differences in the MT-SAT data. As I mentioned in minor comment 7, there are two satellite retrieval algorithms separately for daytime and nighttime. My previous experience working with these datasets are they do not necessarily provide consistent cloud-top height retrievals when switching from one to another. Cloud-top heights from the same cloud systems can differ as much as several kilometers between the two estimates, and during twilight hours (+/- 1-2 hour) when the solar zenith angle is high, the retrievals uncertainties are very large. Did you 1) compare daytime vs. nighttime separately? 2) exclude twilight hours?

**We had not originally done these separations. We went ahead and separated Figure 3c by retrievals made in the daytime and excluded twilight and placed them in the figures below. Our results are insensitive to the time of day.**

[Figure]

**Frequency histogram of VISST retrieved cloud top heights versus CPOL retrieved ETHs using the σ during the daytime (600 to 1700 local time).**

[Figure]

**As above, but during night time**

[Figure]

**As above, but with twilight hours excluded**

8. Why do you choose such a short period of only two months during the peak monsoon TWP-ICE period for a comparison? Particularly when most of the precipitating clouds are deeper than 7-8 km. MTSAT retrievals at this location are available for multiple years in the ARM data archive. Further, wouldn't a more direct comparison be made between CloudSat measured cloud-top heights as it is active remote sensing? For such a long CPOL record and decent spatial coverage, there should be plenty of samples to compare. One of the coauthors of the study (Alain Protat) has made much harder comparisons between CloudSat and ARM cloud radars before (Protat et al. 2014), what stop you from doing that? I understand it takes some effort to do that, but if the paper wants to claim that this new method of estimate ETH from CPOL is closer

to actual cloud-top height,  then a more stringent evaluation is needed than what is presented here.

**We had chosen the two month period during TWP-ICE because the MTSAT data in the ARM archive initially because we, although now we have extended the comparison from 2006 until 2010 where the version 4 MTSAT data are available. We decided to choose this time period because as same version of the VISST product was available to ensure that differences in processing between the differing versions of the VISST product did not interfere with the comparison. Therefore, we now have done a more comprehensive comparison of the ETHs with those from MTSAT than what was done before.**

**Using CloudSat data instead of MTSAT data for the comparison provides us with even fewer data points. Figures R1 and R2 show the derived from the CloudSat calculated using the highest point where the echoes are classified as "good" in the Level 2 compared to those derived using Z < 5 dBZ (Figure R1) and $\sigma < 3$. The gate from CloudSat was compared to the nearest grid point in the CPOL data for the comparison. While there is CloudSat data present from the years 2006 to 2017, since the comparisons are limited to the areas scanned by the CloudSat granules, we only have 2,387 datapoints for comparison compared to having . Therefore, the issue of there being relatively few samples also applies to the CloudSat data. Therefore, we elect to use the MTSAT data in the comparison, as there is actually more data in the two month period we analyzed than in all of the CloudSat granules.**

[Figure]

**(left) The 2D frequency distribution of ETH derived from CPOL using the highest gate where $\sigma < 3$. compared against ETH derived from CloudSat. (right) as left, but using the highest gate where Z > 5 dBZ.**

9.  Page  5  lines  26-29,  if  the  new  technique  using
Σ to  calculate  ETH  (which  I  do not understand, see major comment 4) gives a similar result with  a much simpler Z threshold  approach,  what  is  so  unique  about  this  new  approach then  if  it  does  not perform better than existing ones? Why make a big deal about the novelty?

**This new approach has the major advantage in that, one the noise floor is automatically detected, and, two, second trip echoes are removed. Also, the approach that uses velocity texture is immune to radar miscalibration and therefore can be applied more readily to radar datasets than techniques that use reflectivity. Further, since 5 dBZ can be well above the noise floor of the**

**In this particular case, we have shown that there is little sensitivity in the retrieved ETHs to whether or not one uses reflectivity or velocity texture. Therefore, we have reframed the comparison as a sensitivity test and removed claims in the paper about the novelty of the approach. Rather, we now claim that our ETH retrieval is robust given how little sensitivity there is.**

10. Page 6 line 30, I cannot tell if there is a significant difference in dew point temperature between monsoon break and active period from Fig. 3-4 as they look very similar. You should also compare specific and relative humidity profiles. Mid-level humidity in the tropics is particularly important for supporting deep convection (e.g. Hagos et al. 2014). Showing the difference between the profiles may be useful. Also, given the small difference in the mean thermodynamic profiles between MJO phases, showing a difference in the mean and the variability is useful as well. Please also include the number of soundings that go into the composite.

**We thank the reviewer for this very useful comment. We have added specific humidity profiles to Figures 4 and 5 as well as added the amount of soundings that we derived the profiles from in the figure captions. Given concerns from the other reviewer regarding there being too many variables to look at in the paper, we did not add relative humidity to Figures 4 and 5.**

**The specific humidity at the mid levels is about 1 g/kg higher when the MJO is active and during monsoon conditions. The Hagos et al. (2014) study would suggest that enhanced mid level moisture facilitates the transition from shallow to deep convection, which is consistent with the relative unimodality and the relative unimodality we see in MJO-active/monsoon conditions. Furthermore, the new analysis now shows a greater variability in the winds and specific humidity during MJO inactive/break conditions, which makes for**

11. Figure 6, can you comment on why in this study the "overshooting" mode found in Kumar et al. (2013) is not visible in the much longer dataset? Their study showed that the overshooting mode correspond to intense low-level reflectivity (and inferred larger and more numerous raindrop particles), which tend to occur more during the monsoon break period. Does the 14 km peak in break period (Fig. 6 MJO=1,2) correspond to that mode?

**With the new processing as suggested by previous comments from this reviewer, the 14 km peak has disappeared. However, we have added discussion on the presence of ETH greater than 15 km, which are only present during break conditions.**

12. The congestus mode in Kumar et al. (2013) is defined as ETH < 6.5 km, where in this study it is 8 km. That is not a small difference. 8 km is also significantly higher than the 0C ( 4.5-5.0 km) level where above which freezing and additional latent heating acceleration of vertical motion can occur.   Can you comment on why you choose a larger ETH value for congestus?

**We had originally defined the congestus and deep modes corresponding to the $\mu_1$ and $\mu_2$ derived from the bimodal Gaussian fits which resulted in larger values than what is typically considered congestus. In order to be more consistent with the past literature, we still base our classification of congestus and deep convection from the fits, but now place stricter criteria on what is classified as "congestus" versus "deep convective:"**
   1. **If the ETH distribution is bimodal (0.1 < A < 0.9) then mode 1 is the congestus mode and, mode 2 is the deep convective mode**
   2. **If the ETH distribution is unimodal (A < 0.1) and $\mu_2$ < 6.5 km then the single mode is the congestus mode, otherwise the single mode is the deep convective mode**
   3. **If the ETH distribution is unimodal (A > 0.9) and $\mu_1$ < 6.5 km then the single mode is the congestus mode, otherwise the single mode is the deep convective mode**

13.  Page 8 line 16-18, I thought A is the contribution of mode 1 (congestus), when A increase to 0.9 during active phases of the MJO under monsoon conditions, doesn't that mean most of the convection are congestus, as opposed to deep convection stated in this sentence? That is contrary to the statement of mostly widespread MCSs during active MJO and monsoon.  I think the issue here is using only echo-top height to indicate congestus vs. deep convection is too simplistic. The wide spread MCSs are likely associated with much larger but not as deep convective cells compare to more isolated but deep convection during break period.  One very important indicator for organized convection, i.e., the size of convective cells is ignored in this study.  Larger convective cells and larger updrafts (i.e. in MCSs) carry a majority of the mass flux as reported in both observational analyses (Kumar et al.  2015, Masunaga and Luo 2016) and high-resolution model results (Hagos et al.  2018).  The size of convective cells is just as important (if not more) as the depth of the convective cells. It can be easily quantified with the CPOL data and I think would be a useful quantity to investigate as functions of large-scale regimes along with the ETH analyses.

**We thank the reviewer for this helpful comment. We have taken care to more carefully define congestus and deep convection in response to this reviewer's major comment number 12. Doing this has improved the presentation of the results in this section and reduces confusion.**

[Figure]

We now discuss the convective areas as a function of large scale forcing, noting that there are generally lower convective areas in monsoon or active MJO conditions, which would be consistent with generally weaker convection being present during these conditions. We use this analysis, in combination with the quantitative MCS analysis

**suggested by the reviewer to show that monsoonal conditions are characterized by weaker, but more frequent MCSes.**

14.  Page 8 line 29-30, I think this is an interesting and important finding.  It would be useful to discuss what aspects of the large-scale conditions can help explain larger difference under active MJO (i.e. going back to Fig. 4-5, see major comment 10 about quantifying their relative environment profile difference between active/suppressed MJO, monsoon/break).

**Thank you. We have added a couple of sentences of discussion here on how the differences in the environmental profiles observed between MJO phases could be contributing to the differences in the ETHs seen:**
**"***Considering that, in Figures 4a and 5a show greater increases in equivalent potential temperature with height above the 5 km stable layer during MJO inactive conditions, this suggests that the midlevel thermodynamic profiles support greater inhibition of the convection in the deep convective mode when the active phase of the MJO is away from Australia.***"**

**With the new ETH processing and more careful definition of the modes, we also now observe that the congestus mode is more sensitive to the presence of the monsoon while the deep convective mode is more sensitive to the presence of the MJO**

15.  Figure 9,  other than the more obvious enhanced frequency of deep convective during daytime, the difference for the rest of the panels between MJO phases are difficult to see. Perhaps adding a difference panel would help.

**We attempted to add a difference panel to Figure 9, but this made the figure too busy to be readable. Given that the other reviewer commented that there were too many variables in the paper, we did not add a difference panel here.**

16. Page 9 line 26-29, why do you have to guess that the enhanced nighttime peak over the ocean during monsoon period is due to MJO? It is relatively easy to identify MCSs in CPOL data (e.g., Rowe et al.  2014 used a simple criteria of precipitation feature major axis length > 100 km to identify MCS in ground-based radar observations), why not actually quantify MCS frequency changes to better support your claim?

**We have quantified the radar coverage of MCSes using the methodology of Rowe and Houze (2014) for each scan as the reviewer suggested and have added the average number of MCSes in the radar domain per scan for a given large scale forcing regime and time of day in Table 1. The normalization was done to ensure that differences in the number of MCSes identified was not due to the differing lengths of time spent in each regime.**

**The results in Table 1 clearly show that, during both an active MJO and active monsoon, that on average more MCSes are present in the radar domain during these conditions. On average, there is also increased presence of MCSes during the daytime compared to night time. Therefore, doing a quantitative analysis suggests that, in most conditions except active monsoon, there are more MCSes at night than during the day. Therefore, most of the conclusions we had before still hold, and where any changes had to be made the discussion was changed accordingly. We also now refer to the frequencies in Table 1 in addition to the already provided references to justify our claims of MCS frequency in Section 4.**

17. Figures 10-11, given the strong diurnal cycle between land vs. oceanic area shown in Figures 8-9, did you separate land area and ocean area when calculating their ETH occurrence? I also suggest plotting Figures 10-11 in local time to make it easier for Readers.

**We did not initially separate the diurnal cycle figures by land and ocean. At the suggestion of the other reviewer who was concerned about too many variables being presented in this section we have not added an extra figure separating the diurnal cycle by land and ocean.**
**Figures 10 and 11 are now plotted as a function of local time.**

18. Page 11 line 13-14, which figure shows a peak of ETH around 5-6 km during break conditions? Figure 6 when MJO is away from Australia (phases 1-2) generally show a peak between 6-8 km, but drops to 4 km in phases 3-4.

**We have modified this conclusion to be more consistent with the analysis in Figure 6, which has changed with the ETH reprocessing.**

Minor Comments
1. Page 1 line 2, technically validation of convective processes in GCMs do not "require" such statistics, perhaps it's better to say "could benefit from" such statistics.

**We have changed the wording in this sentence to the suggested wording from the reviewer.**

2. Page 1 line 5, why does it have to be for a specific model? These observations can be useful for any large-scale model validations.

**We were focusing on E3SM as for that model there is an undergoing development to have the model compute results at a 12 km resolution that is high enough such that both MCSes are resolves and the assumptions made in convective parameterizations may not apply. However, since we agree that this is really useful for any GCM, we have changed this sentence to state that this dataset is useful for the validation of convective**

**processes in any GCM.**

3.  Page 1 line 22, Jensen et al.  (1994) stated the 100 W/m2 is solar forcing, not net radiative forcing, please clarify that in the statement.

**We have corrected this statement.**

4. Page 2 line 19, "...of of an intense..."

**We have removed the extra "of."**

5. Page 2 line 27, "...for one wet season in and found..." in what?

**We have completed this sentence**

6. Page 4 line 1, spell out the acronym "ACRF".

**Due to a recent change in the name of the ARM Climate Research Facility (ACRF) to ARM Facility, we have changed this to say "ARM Facility."**

7.   Page 4 line 4-6,  I believe the MTSAT data,  at least the inferred channel spatial resolution should be  5 km, not 1 km. Also, the VISST technique uses all available geo-stationary satellite channels, including visible, water vapor, near IR, and IR channels to retrieve cloud properties during day time.  For nighttime, a different technique SIST is used for the lack of visible channel.

**We now mention that the VISST technique uses these two techniques in this sentence. Also, according to the ARM archive, the resolution is 4 km, which we now mention here.**

8.  Page 5 line 14, you mentioned "normalized frequency distribution", what is the unit of the shading in Figure 3? The large numbers appears to be just a count, if so it is not normalized frequency.

**We have removed the word "normalized" from this sentence.**

9.  Page 5 line 31, do not use "cloud top height" and ETH interchangeably.  You have not established in this study that CPOL ETH is equivalent to cloud top height.

**We have changed this phrase to say "echo top height."**

10. Figs. 3-4, why is the vertical scale different between the two figures?

**The vertical scales are all now the same between Figures 3 and 4.**

11. Page 7 line 12, spell out p.d.f. in section headings.

**We have corrected this to say "normalized frequency distributions" which is actually what is being plotted.**

12. Page 8 line 8, "In 7,..." do you mean in "Eq. (1)"?

**We meant to say Figure 7. It says so now.**

**References for Authors' Response**
Barnes, S. L., A technique for maximizing details in numerical weather-map analysis. Journal of Applied Meteorology. 3,4,: 396–409, doi:10.1175/1520-0450(1964)003<0396:ATFMDI>2.0.CO;2, 1964

Del Genio, A.D., Representing the Sensitivity of Convective Cloud Systems to Tropospheric Humidity in General Circulation Models, *Surv. Geophys*, 33: 637. https://doi.org/10.1007/s10712-011-9148-9, 2012.

References
Protat, A., S.A. Young, S.A. McFarlane, T. L'Ecuyer, G.G. Mace, J.M. Comstock, C.N. Long, E. Berry, and J. Delanoë, 2014: Reconciling Ground-Based and Space-Based Estimates of the Frequency of Occurrence and Radiative Effect of Clouds around Darwin, Australia. J. Appl. Meteor. Climatol., 53, 456–478, https://doi.org/10.1175/JAMC-Kumar, V. V., C. Jakob, A. Protat, C. R. Williams, and P. T. May (2015), characteristics of tropical cumulus clouds from wind profiler observations at Darwin,
Australia, J. Atmos. Sci., 72, 1837–1855, doi:10.1175/JAS-D-14-0259.1.
Masunaga, H., and Z. J. Luo (2016), Convective and large scale mass flux profiles over tropical oceans determined from synergistic analysis of a suite of satellite observations, J. Geophys. Res. Atmos., 121, 7958–7974, doi: 10.1002/2016JD024753.
Hagos, S., Z. Feng, K. Landu, and C. N. Long (2014), Advection, moistening, and shallow-to-deep convection transitions during the initiation and propagation of Madden-Julian Oscillation, J. Adv. Model. Earth Syst., 06, doi:10.1002/2014MS000335.
Hagos, S., Feng, Z., Plant, R. S., Houze, R. A., Xiao, H. (2018). A stochastic framework for modeling the population dynamics of convective clouds. Journal of Advances in Modeling Earth Systems, 10. https://doi.org/10.1002/2017MS001214.
Rowe, A. K., and R. A. Houze Jr. (2014), Microphysical characteristics of MJO convection over the Indian Ocean during DYNAMO, J. Geophys. Res. Atmos., 119, 2543–2554, doi: 10.1002/2013JD020799.

---

## Author Comment (AC2) · 21 Sep 2018

General Comments This study aims to define characteristics of monsoon phase (break vs active) within context of MJO phase over Australia for northern AU region. 17 years of radar data are used to increase sample size and develop statically significant results. Overall I think this is an interesting study that shows the impact of MJO phase on cloud top heights for active and inactive periods drawing on precipitation and thermodynamic characteristics to explain the results.

**We would like to give thanks to the reviewer for taking the time to provide careful feedback on the manuscript. We agree that the results from the long term dataset presented in the paper are an interesting addition to the literature. In response to another reviewer we have re-proccessed the ETH to be defined as the lowest gate in the column where the velocity texture is greater than 3, when previously it was the highest gate in the column where it was less than 3 (not mentioned in the draft manuscript). We have also excluded data greater than 100 km in range This was done to account for regions where a detached anvil or cirrus cloud layer was above the precipitating convection and helps to ensure that we are only including the precipitation convection in our analysis instead of remnant anvils. Furthermore, it was found that the year 2013 was missing from the ETH database, so now data from the year 2013 are included. Therefore, the updated processing of the ETHs has resulted in some changes to the conclusions of this study.**

**In response to another reviewer we have also added an analysis of the convective areas and quantified the number of MCSs using the technique used by Rowe and Houze (2014). We felt that now that this paper covers more than just the ETHs, we have changed the title to "*A 17 year climatology of the macrophysical properties of convection in Darwin.*"**

My major criticism is that the description is confusing and hard to follow in parts of the statistical analysis (Sec. 4.1) and diurnal cycle (Sec. 4.2) sections

as described in the specific comments below. The confusion is due to 1) combination of too many variables to consider when trying to correlate interpretations stated in the text to results shown in selected figures: active vs inactive monsoon, MJO phase, day vs night, ocean vs land; and 2) the text is not always explicit in terms of which panel of a figure is being used to advance the argument. In consequence, I as the reader, have sometimes come to a different conclusion when interpreting the figures in question compared to the authors. I have also noted these instances below.

**We agree that there are quite a few variables that we are stratifying the ETHs and convective areas against and it can be quite overwhelming to the reader. However, the macrophysical properties of convection in Darwin are sensitive to numerous factors including the time of day, the phase of the MJO or the monsoon and even the surface characteristics. We strongly feel that it is important to show how the ETHs can vary as as a function of such characteristics.**

**Since we are unable to outright eliminate any independent variables from our analysis, we have taken some steps to simplify the presentation of the data to make it easier to interpret:**

1. **We now use a more rigorous definition of "congestus" versus "deep convection" that is based off of the threshold used by Kumar et al. (2013), who defined the boundary between congestus and deep convection to be 6.5 km. This, not only being more rigorous and consistent with past literature, also makes the results in Section 4.1 and 4.2 easier to read.**
2. **Every multipanel figure are now labelled by letter and each subpanel is now referred to explicitly.**
3. **The figures showing the ETH distributions as a function of time now just show the frequency distributions with time, reducing the number of figures by 1 and cleaning up clutter. This was done as there was little discussion and few conclusions to be made about how bimodality varies throughout the day.**

**Furthermore, the other reviewer requested a quantification of MCSes as well as an an analysis of convective areas in order to more adequately characterize the presence of deep convection than what was done in this manuscript. We have added such analysis to the statistical analysis section.**

A more modest critique concerns the spectrum width thresholding technique used to discriminate echo top height as opposed to a minimum reflectivity threshold. On the bright side, the method appears to work reasonably well. However, the end results is that there does not seem to be any real difference in the results when compared to simply applying a minimum reflectivity threshold (which is the traditional approach)
so I am left scratching my head when trying to understand the real advantage of the methodology.

**We do not use spectrum width in order to determine the echo top height, but rather, we use the texture of the radial velocity field. This is the standard deviation of the radial velocity of a 3 by 3 gate window surrounding a gate and not the standard deviation of the Doppler spectrum. Radars like CPOL that have a phase that varies randomly from pulse to pulse produce radial velocities that vary randomly gate-to-gate when there is no single scattering returned radiation.**

**The advantage of such an approach is that (1) the noise floor is automatically detected, (2) we can potentially be able to keep regions in cloud where Z is lower than 5 dBZ, which would be more frequent with 50 km range of CPOL. Finally, this methodology is immune to radar miscalibration and less sensitive to attenuation. This would give us, in theory, more representative of the true cloud top height. While we have removed the "novel" and "new" wording in the abstract, we now more explicitly list these advantages in the introduction and frame the discussion as a sensitivity test. We also have improved the explanation of the velocity texture based ETH retrieval technique so that it is easier to understand both how it works and its potential advantages.**

**While, for this particular dataset, we arrive at the null conclusion that the ETHs retrieved using the velocity texture methodology and using reflectivity are comparable. We feel that the inclusion of null results in a paper is something that should be done more to guide future research on what methodologies have been tried. This is not done enough in papers in our opinion, and given that this section is short, adding a null result does not significantly lengthen the paper.**

Specific Comments 1. P. 4, line 12; please define gate spacing and resolution

**We have added the 300 m resolution and gate spacing to this section.**

2. P. 5, line 7: What is the spatial resolution of the satellite data? If it's less than radar resolution it's not clear what a relative comparison tells us regarding the performance of the radar-based ETH algorithm.

**The spatial resolution of the satellite data at 4 km, so interpolation of the satellite data to the radar's grid was needed.**

**In response to the other reviewer, we have expanded the analysis in this section to 4 years of MTSAT data from 2006 to 2010, using version 4 of the VISST product. While we acknowledge that the coarser resolution of the satellite introduces uncertainties into the satellite retrieval, over time scales of years the relative seasonal variability in cloud top heights should be captured.**

3. P. 5, line 15: similar to previous comment - to understand the differences in cpol vs satellite – what is satellite brightness temp keying off of – what depth of cloud is considered?

**The VISST technique uses both the solar and infrared channels at multiple wavelengths to retrieve the cloud properties. According to Cheng et al. (2010), the retrieval is keying off of a height "somewhere**

below physical cloud top," but they do not quantify exactly where. We have chosen not to attempt to quantify this since this is extremely difficult to do.

4. P. 5, line 17: cc of 0.49 is not very good
**This, while weak, is a statistically significant correlation according to a chi-squared test. The mention of the statistical test has been added to this sentence.**

5. P. 5, line 20: this statement assumes the satellite is capturing the variability …

**Over timescales of seasons and spatial scales of hundreds of kilometers, we would fully expect this to be the case.**

6. Fig. 3 please state in the caption what the color shading represents

**We now state that the shading represents the number of counts.**

7. P 6. Lines 25-30: In references to Figs 4-5, seems like the big differences are between monsoon phase instead of MJO phase?

**For the winds, this is the case. We have noted that there is little difference between the wind speeds between active and inactive MJO conditions.**

8. P. 7, line 7 – There are several other older references
that show this behavior: Cifelli and Rutledge 1994 (JAS); 1998 (QJRMS)

**We have added these references to this sentence.**

9. P. 7, line 27-28: some hint of trade wind layer in MJO=3 for break (Fig. 6)?

**We have now changed this sentence to acknowledge that we could be sensing some of the trade wind layer during break conditions, but we ultimately need a radar that is sensitive to cloud particles to characterize it:**

**"***Also, some evidence of the trade wind mode is visible in Figures 6a-h. However, since the 2 km modes in Johnson et al. (1999) and Kumar et al. (2013) were observed using measurements with a cloud radar that would be more sensitive to liquid cloud droplets than CPOL, more sufficient quantification of this mode would require a radar with a lower minimum discernable signal than CPOL.***"**

10. P. 8, line 8: This is a minor point but it should be noted that the heights of the different modes that are stated here are approximate. For example, in Fig. 7c the height of mode 2 does not appear to actually reach 15 km
...
**We have added the word "approximately" before the quantities in this sentence, and fixed the faulty reference to Figure 7.**

11. P. 8, line 12-18: there is some confusion looking
at Fig 7. My read of the red line (A=congestus) in MJO phases 4-7 is
~0.05 – 0.6 for break (Fig 7b) – not 0.8-0.5 as described - and
~0.1-0.4 for monsoon (Fig. 7d) – not > 0.9 as described. Also, the statement on line 14 about unimodality is confusing:

Fig 7b,d show that there is a significant contribution from the congestus mode in break conditions while the MJO is over AU (Fig. 7b, MJO phases 6,7). Similar in monsoon conditions for MJO phase 6 – see Fig 7d. I think the confusion noted above could be avoided by stating more clearly which features in specific figure panels are being referred to.

**We have made a more rigorous definition of "congestus" versus "deep" convective modes that makes this section easier to understand. We**

**determine whether the modes present are congestus or deep convection based on the average location of the modes:**

1. **If the ETH distribution is bimodal (0.1 < A < 0.9) then mode 1 is the congestus mode and, mode 2 is the deep convective mode**
2. **If the ETH distribution is unimodal (A < 0.1) and $\mu_2$ < 6.5 km then the single mode is the congestus mode, otherwise the single mode is the deep convective mode**
3. **If the ETH distribution is unimodal (A > 0.9) and $\mu_1$ < 6.5 km then the single mode is the congestus mode, otherwise the single mode is the deep convective mode**

**The 6.5 km threshold was chosen based off of Kumar et al. (2013).**

**We agree, the statement about the unimodality was confusing. We removed it.**

12. Fig. 8 – please state in the caption and the figure that this is for break conditions

**We have added this information to the caption.**

13. P. 9 -please call out panels explicitly in reference to Figs. 8-9

**Labels have been added to these panels and now the discussion refers to each relevant panel explicitly.**

14. The discussion jumps to Fig 10 before discussing Fig. 9

**This is no longer the case.**

15. P. 9, line 24: which panel of Fig 10? My read of comparing Fig 10 a and Fig 10b is that during the day there is a higher frequency of deep convection when the MJO is over AU (assume that includes Tiwi islands as well) compared to when MJO is elsewhere.

**We have added a reference to Figure 10d.**

16. P. 9, 25-26: I don't understand the point about what is being extended in this study vs previous work.

**We agree, this sentence was quite confusing. We have rephrased this sentence from:**

"*Figure 10d shows a greater frequency of deep convective ETHs over the Tiwi Islands when the MJO is inactive over Australia during the day, which is consistent with increased rainfall over this region.*"

17. P. 10 lines 11-12 – where do the number of days come from?

**The number of days comes from the sounding classification in Section 3.3**

**References:**

Chang, F.-L., P. Minnis, J. K. Ayers, M. J. McGill, R. Palikonda, D. A. Spangenberg, W. L. Smith Jr., and C. R. Yost (2010), Evaluation of satellite-based upper troposphere cloud top height retrievals in multilayer cloud conditions during TC4, *J. Geophys. Res.*, 115, D00J05, doi: 10.1029/2009JD013305.

---

## Referee Report (RR1)

**Review:** ACP-2018-408-R1

Title: A 17 year climatology of the macrophysical properties of convection in Darwin

**Recommendation: Minor Revision.**

**Summary**

The authors have done extensive revision to address many of my comments. I particularly commend the authors to extend their analysis beyond just using the vertical height of convection, but also look into the horizontal dimension of convection, including identifying MCSs explicitly. The added results on convective cell sizes and MCSs nicely complement the ETH analysis of the study. As a result, I think the paper has improved significantly compared to the original manuscript.

There are a few places with some relatively minor remaining issues that should be revised/clarified before the paper is published. Please see my comments in orange below. I recommend minor revision.

**Comments**

Page 6 line 9-12, actually the daytime satellite retrieval algorithm is referred to as VISST, and the nighttime algorithm is referred to as SIST: https://cloudsway2.larc.nasa.gov (click the VISST/SIST link).

7. There is also the issue of daytime vs. nighttime retrieval differences in the MTSAT data. …

We had not originally done these separations. We went ahead and separated Figure 3c by retrievals made in the daytime and excluded twilight and placed them in the figures below. Our results are insensitive to the time of day.

Thank you for attempting to separately compare CPOL and MTSAT echo/cloud-top height retrievals between day vs. night. It's good to know that the results are insensitive to the two different algorithms, suggesting at least that the VISST/SIST provide statistically consistent convective cloud-top height retrievals. More importantly, comparing your original Figure 3 with the new one where the CPOL ETH is now defined as the lowest precipitating convective echo-top, the mean differences as well as the spread is now significantly larger (i.e., CPOL ETH is now significantly lower than MTSAT for cloud-tops above 7.5 km. That means the occurrence of multi-layer clouds above precipitating convective cells are indeed quite frequent, obviously the passive satellite only "sees" the highest layer cloud tops for these optically thick clouds, while the CPOL could detect distinct layers. I think this is a useful result to point out, perhaps you could add a couple sentences describing the difference when comparing CPOL lowest layer ETH vs. max ETH.

Figure 4. I don't think the temperature profile (Fig. 4b) provides much useful information. Similarly, dew point temperature and specific humidity is also somewhat duplicative. You also did not specifically discuss these quantities in the text. Relative

humidity is a more useful quantity that differentiates probability of transitioning to deep convection. I suggest you could simplify Fig. 4, 5 with just 4 panels, theta E, RH, U, V.

Page 7 line 33-34, "This shows that there are a greater number of cases with westerly flow advecting moisture from the Indian Ocean when the MJO is active over Australia." I think the 95[th] percentile U wind being larger when MJO is over Australia in Fig. 4e, 5e only means the tail of the zonal wind is stronger. That does not necessarily mean moisture advection is larger. You could have stronger zonal wind with drier air. What would support your claim is to compare moisture flux (U * qv) profiles.

Figure 7d, what concerns me is the complete reverse between 100% deep convection in MJO phase 1 to 100% congestus in MJO phase 2. Is it reasonable to believe that: 1) 100% of convective clouds during when MJO is away from Australia are all deep convection (or congestus)? 2) All of the deep convective clouds suddenly all changed to congestus in the next MJO phase? You also ignore this figure in the text when discussing Fig. 7 on page 10. I don't think I understand what u1 and u2 mean, and how are the fractional contribution of modes in Fig. 7b,d are calculated.

I find the new Figure 8 quite interesting. Thank you for adding this analysis. The result suggests that when MJO is over Australia, monsoon/break has no effect for the population of convective cells. But when MJO is away from Australia, monsoon periods have narrower distribution of convective cell sizes, and also less frequent large convective cells than break period. I think this could be one of the highlighted results of the paper.

Table 1 caption is misleading, the numbers in the table are frequencies, not "average number of MCSs". The caption also has some missing information: "… using the criteria of (?)".

Page 11 line 15-16, Table 1 only provides MCS frequencies separately for break/monsoon, and MJO away/over Australia (unless I misunderstood). But it does not provide frequencies during break+MJO away from Australia, break+MJO over Australia, monsoon+MJO away, monsoon+MJO over. So it is inconsistent with the discussions in Fig. 7 in this paragraph, which combines the monsoon and MJO conditions. How about adding the MCS frequency calculations for these combined conditions in the table to align better with the 4 key large-scale conditions of the paper?

Page 13 line 12-14, "Since MCSs are larger …,  the reduced frequency of MCSs in the inactive phase of the MJO in Table 1." The last part of the sentence seems to have some missing words.

---

## Editor Decision (ED1)

In response to the two reviewer's helpful comments, we have improved the manuscript in various ways. The most major changes to the manuscript:

1. Language regarding the novelty of the velocity texture method for retrieving echo top height (ETH) has been removed from the abstract and potential advantages of such a methodology are now more explicitly delineated. In addition, we have clarified how we are deriving the ETH.

2. Our comparison of the ETHs against the MTSAT data has now been extended to the 2006-2011 timeframe when the VISST version 2 data were available.

3. All multipanel figures are now labelled by letter for each panel for easier readability.

4. The ETHs were reprocessed to account for cloud layers above the precipitating convection. This resulted in some changes to the conclusions.

5. The amount of subpanels in the section analyzing the diurnal cycle have now been reduced for easier readability as some panels did not add to the manuscript.

6. We have now more explicitly defined our definition of "congestus" and "deep convection" in a manner that should be both easier to interpret and is consistent with what has been done in past literature.

7. We have added an objective analysis of the number of MCSes and also looked at how the convective areas vary in the differing large scale forcing regimes.

8. We have more explicitly linked how the differences in the thermodynamic profiles between an active and inactive MJO can contribute to the differences in the ETHs observed between an active and inactive MJO.

More minor changes to the manuscript are detailed in our response to the reviewers' comments. We again thank the two reviewers for taking the time to provide helpful comments.

Review
: ACP-2018-408
Title: A 17 year climatology of convective cloud top heights in Darwin
Recommendation: Major Revision.

Summary
This study examines the variability of convective echo-top heights (ETH) observed by long-term CPOL radar in Darwin Australia as functions of large-scale conditions during active/suppressed MJO and monsoon/break periods. A new technique to estimate ETH is described and compared to a traditional reflectivity threshold based technique, and to a short period of geostationary satellite retrieved cloud-top heights. The study then continues to partition ETH distributions by combining MJO phases with monsoon indices, and concludes that MJO has relatively stronger influence to convective ETH than monsoon.

In general, I think there are valuable results presented in this paper, particularly illustrating the MJO activity over Australia are relatively more important in regulating convective ETH compared to monsoon vs. break conditions, which previous studies have not investigated. However, there are many aspects of the study that need improvement to make it a significant contribution worthy of publication in ACP.

First, the "novel new technique" described in this study does not prove to be any better (or even different) than simpler existing method in calculating ETH, at least based on the short and in my opinion problematic comparison with a passive satellite cloud-top height retrieval dataset. Then why make a big deal about it? There is nothing wrong with using existing ETH technique, especially if you can't show that this new method works better than previous ones.

**We would like to thank the reviewer for their insightful comments on this manuscript. In a phase randomized radar significant returns have a "Smooth" radial velocity while regions of no return or multi-path have radial velocity that varies randomly from -nyquist to +nyquist. This allows texture of radial velocity to provide a very "clean" determination of a significant return. Most importantly it provides a consistent definition of echo top height rather than relying on an arbitrary threshold. Texture allows the echo top height to match the height of minimum detectable signal.**
**We have rephrased the "novel new technique" wording in the abstract to simply state that we are assessing the applicability of such a methodology. This therefore reduces the scope of this section from attempting to state which methodology better estimates the cloud top height to simply a comparison that assesses the sensitivity of our results to the ETH retrieval technique used. We still compare the retrieved ETHs against cloud top heights estimated from satellites in order to assess whether or not the retrieval can capture the statistical variability in ETHs without necessarily knowing the uncertainty. Given a dataset as large as this, this approach is feasible as this tells us that we can observe the relative interseasonal variability in ETHs. Furthermore, we feel that it is**

**important to show the sensitivity of the ETHs to the retrieval technique used, which few studies do, even if a null result is shown.**

**Given the revision in scope of Section 3, the new wording in the abstract rephases the testing to demonstrate that there is little sensitivity to the ETH with retrieval technique and that our retrieval captures the relative seasonal variability in cloud top height:**

*"Retrieved ETHs are correlated with those from MTSAT retrieved cloud top heights, showing that the ETHs capture the relative variability in cloud top heights over seasonal scales. "*

**The wording in Section 3 regarding the methodology comparison already stated that both methods were roughly equivalent, so no changes were made there. However, since both methodologies give similar echo top heights as shown in the comparison in Section 3, we still elect to use the velocity texture methodology since it provides comparable results to using reflectivity.**

Second, I think the author need to connect the relative difference in large-scale conditions between active/suppressed MJO and monsoon/break to help explain why MJO has stronger modulation to convective ETH. Looking more at the difference in variability of the sounding profiles (rather than just the mean) between those conditions may be useful.

**We have added 5th and 95th percentiles to Figures 4 and 5 to represent the variability seen in each of the large scale forcing regimes and also now add specific humidity. We also now better connect the differences between the large scale forcings in the discussion of Figures 4 and 5 by examining the differences in the distributions of winds and specific humidities, with two paragraphs in Section 3.3 instead of one demonstrating the relative differences between the regimes. In it is shown in this section now that:**
1. **There is a greater variability in the Surface to 500 hPa winds during an inactive MJO, which contributes to fewer cases where stronger flow of moisture from the west are occurring.**
2. **There is reduced mid-level moisture (4 to 8 km) during inactive MJO/break conditions, as well as wider variability of such moisture as indicated by the specific humidity profiles. This suggests that the large scale environment during an active MJO is more favorable for the transition of congestus to deep convection.**

Third, the study misses an opportunity to examine how do these different large-scale regimes modulate an important aspect of convection in Darwin: the variability of convective cell sizes. Several recent studies have pointed out the importance of convective cell sizes to mass flux, a critical aspect to cumulus parameterizations. Also, analyses of spatial scales of convection would provide more concrete conclusions on changes

of MCS activities with large-scale regimes, which the paper makes many reference but did not show supporting evidence.

**We have added an analysis of both how the cell sizes vary and now objectively quantify the number of MCSes using Rowe and Houze (2014)'s methodology to justify our claims about MCS coverage. In addition we have added material to the introduction now introducing why cell size is important and what past studies have concluded about how the cell sizes in Darwin vary in differing large scale forcings in the introduction. We thank the reviewer for this suggestion!**

I provided more detail comments below on various places in the paper that need improvement. Because I do see the value of the long-term tropical radar observations, I recommend major revision of the paper before it can be accepted for publication.

Major Comments
1. Page 2 line 20, please be more specific on what aspects of convective parameterizations are "poor". Poor in doing what?

**We have replaced this sentence to be more specific by what we mean by "convective parameterizations are poor:"**

*Also, convective parameterizations in GCMs do not account for mesoscale organization resulting in insufficient sensitivity to upper tropospheric humidity (Del Genio, 2012).*

2. Section 3.1, what kind of quality control procedures were applied to the raw radial radar data? How do you handle ground clutter, AP, and noise that are particularly prevalent in CPOL data at lower level, which could affect your ETH estimates?

**The great advantage of the use of velocity texture compared to using reflectivity is that the noise floor can be detected automatically since we expect the velocity field in regions of noise and second trip echoes to be random, resulting in higher velocity textures. We are also doing the following steps:**

- **Excluding gates with differential reflectivity < -3 and > 7 dB.**
- **Excluding gates with < 0.45 cross-correlation coefficient.**
- **Excluding gates with differential phase texture > 20**
- **Excluding gates with reflectivity less than -20 dBZ and greater than 80 dBZ.**

**We have added these steps into a bulleted list into Section 3.1**

3. Page 4 line 18, why choose a "box" when the radar scans are circular? The diagonal

corners of the box are 140 km away from CPOL, which is much further than 100 km radius where sampling and resolution are better.

**We chose a "box" since we interpolated the radial data onto a Cartesian grid for easier spatial analysis and using boxes is easier with data in Cartesian coordinates. While creating the Cartesian grid, we used Barnes (1964)'s weighting function with a radius of influence that increases with distance from the radar in order to account for the decreased sampling and resolution as a function of distance from the radar. We conducted visual analyses of such grids and had determined that using a 100 km by 100 km box gave reasonable coverage, even at ranges of 140 km.**

4. Page 5 paragraph 2 and 3, I do not understand how is ETH estimated in CPOL using the Doppler velocity standard deviation ($\sigma$) technique. Figure 2 shows example of using $\Sigma > 3$ to remove non-precipitating radar echoes, but then how is ETH determined from the remaining echoes? How is it different from simply just using Z threshold > 5 dBZ?

**Previously, the ETH was determined by looking at the gate above the highest valid gate after the $\sigma$ technique was applied. The ETH is now determined by looking for the first echo in the column that is masked using the $\sigma$ technique. We have added some clarifying wording in this paragraph to explain how we retrieve the ETHs from the masked data: "***The ETHs are then determined by looking at the lowest gate in the column that is masked. We use the lowest gate in the column in order to ensure that we are capturing the ETHs of the precipitating convection, and not that of detrained anvils and cirrus that can lie above the precipitating convection.***"**

**This is different from using Z > 5 dBZ as we are not using reflectivity at all.**

5.  Please also clarify how is ETH estimated using the Z > 5 dBZ threshold method. Do you go from surface upwards and find the first height level where Z drops below 5 dBZ? Or do you go from top downwards to find first level where Z exceeds 5 dBZ (i.e. max height of 5 dBZ in a column)?  They could give very different results because the second approach would get cirrus/anvil clouds that are above precipitating convective cloud-tops (particularly for existence of multi-layer clouds).

**We had originally used the topmost point in the column where Z > 5 dBZ, or the second approach. However, in order to account for cases of multi-layer clouds, we have switched to the first approach as it is more representative of the precipitating convection. We now also make it more clear how we are deriving the ETH from Z in Section 3.**

6.  Figure 3, the authors did not provide enough detail about how the CPOL data and MTSAT data are matched for the comparison.  Given the two datasets have different spatial resolution, do you match them by interpolating one to another?  Do you only compare grid points that both CPOL and MTSAT identified as echo/cloud?

**We interpolated the MTSAT data to CPOL's grid and now say so in Section 3.2. In addition we also state that we only compare points that were identified as in cloud by the VISST product and as convective by the Steiner et al. (1995) algorithm in CPOL.**

*"Since the two datasets are at differing resolution, the MTSAT data are interpolated onto the same grid as the CPOL data for the comparison. Furthermore, to ensure that we are comparing points that are in precipitating convection we both only include points from MTSAT where the VISST product identified cloud and where the convective classification algorithm, detailed in Section 3.4, classified the grid points as precipitating convection."*

7.   There is also the issue of daytime vs.  nighttime retrieval differences in the MT-SAT data.  As I mentioned in minor comment 7, there are two satellite retrieval algorithms separately for daytime and nighttime.  My previous experience working with these datasets are they do not necessarily provide consistent cloud-top height retrievals when switching from one to another.  Cloud-top heights from the same cloud systems can differ as much as several kilometers between the two estimates, and during twilight hours (+/- 1-2 hour) when the solar zenith angle is high, the retrievals uncertainties are very large. Did you 1) compare daytime vs. nighttime separately? 2) exclude twilight hours?

**We had not originally done these separations. We went ahead and separated Figure 3c by retrievals made in the daytime and excluded twilight and placed them in the figures below. Our results are insensitive to the time of day.**

[Figure]

**Frequency histogram of VISST retrieved cloud top heights versus CPOL retrieved ETHs using the $\sigma$ during the daytime (600 to 1700 local time).**

[Figure]

**As above, but during night time**

[Figure]

**As above, but with twilight hours excluded**

8. Why do you choose such a short period of only two months during the peak monsoon TWP-ICE period for a comparison?  Particularly when most of the precipitating clouds are deeper than 7-8 km.  MTSAT retrievals at this location are available for multiple years in the ARM data archive.  Further, wouldn't a more direct comparison be made between CloudSat measured cloud-top heights as it is active remote sensing? For such a long CPOL record and decent spatial coverage, there should be plenty of samples to compare.  One of the coauthors of the study (Alain Protat) has made much harder comparisons between CloudSat and ARM cloud radars before (Protat et al.  2014), what stop you from doing that?   I understand it takes some effort to do that,  but if the paper wants to claim that this new method of estimate ETH from CPOL is closer

to actual cloud-top height,  then a more stringent evaluation is needed than what is presented here.

**We had chosen the two month period during TWP-ICE because the MTSAT data in the ARM archive initially because we, although now we have extended the comparison from 2006 until 2010 where the version 4 MTSAT data are available. We decided to choose this time period because as same version of the VISST product was available to ensure that differences in processing between the differing versions of the VISST product did not interfere with the comparison. Therefore, we now have done a more comprehensive comparison of the ETHs with those from MTSAT than what was done before.**

**Using CloudSat data instead of MTSAT data for the comparison provides us with even fewer data points. Figures R1 and R2 show the derived from the CloudSat calculated using the highest point where the echoes are classified as "good" in the Level 2 compared to those derived using Z < 5 dBZ (Figure R1) and $\sigma < 3$. The gate from CloudSat was compared to the nearest grid point in the CPOL data for the comparison. While there is CloudSat data present from the years 2006 to 2017, since the comparisons are limited to the areas scanned by the CloudSat granules, we only have 2,387 datapoints for comparison compared to having . Therefore, the issue of there being relatively few samples also applies to the CloudSat data. Therefore, we elect to use the MTSAT data in the comparison, as there is actually more data in the two month period we analyzed than in all of the CloudSat granules.**

[Figure]

**(left) The 2D frequency distribution of ETH derived from CPOL using the highest gate where $\sigma < 3$. compared against ETH derived from CloudSat. (right) as left, but using the highest gate where Z > 5 dBZ.**

9.  Page 5 lines 26-29, if the  new  technique  using
Σ to  calculate  ETH  (which  I  do not understand, see major comment 4) gives a similar result with a much simpler Z threshold  approach,  what  is  so  unique  about  this  new  approach then  if  it  does  not perform better than existing ones? Why make a big deal about the novelty?

**This new approach has the major advantage in that, one the noise floor is automatically detected, and, two, second trip echoes are removed. Also, the approach that uses velocity texture is immune to radar miscalibration and therefore can be applied more readily to radar datasets than techniques that use reflectivity. Further, since 5 dBZ can be well above the noise floor of the**

**In this particular case, we have shown that there is little sensitivity in the retrieved ETHs to whether or not one uses reflectivity or velocity texture. Therefore, we have reframed the comparison as a sensitivity test and removed claims in the paper about the novelty of the approach. Rather, we now claim that our ETH retrieval is robust given how little sensitivity there is.**

10. Page 6 line 30, I cannot tell if there is a significant difference in dew point temperature between monsoon break and active period from Fig. 3-4 as they look very similar. You should also compare specific and relative humidity profiles. Mid-level humidity in the tropics is particularly important for supporting deep convection (e.g. Hagos et al. 2014). Showing the difference between the profiles may be useful. Also, given the small difference in the mean thermodynamic profiles between MJO phases, showing a difference in the mean and the variability is useful as well. Please also include the number of soundings that go into the composite.

**We thank the reviewer for this very useful comment. We have added specific humidity profiles to Figures 4 and 5 as well as added the amount of soundings that we derived the profiles from in the figure captions. Given concerns from the other reviewer regarding there being too many variables to look at in the paper, we did not add relative humidity to Figures 4 and 5.**

**The specific humidity at the mid levels is about 1 g/kg higher when the MJO is active and during monsoon conditions. The Hagos et al. (2014) study would suggest that enhanced mid level moisture facilitates the transition from shallow to deep convection, which is consistent with the relative unimodality and the relative unimodality we see in MJO-active/monsoon conditions. Furthermore, the new analysis now shows a greater variability in the winds and specific humidity during MJO inactive/break conditions, which makes for**

11. Figure 6, can you comment on why in this study the "overshooting" mode found in Kumar et al. (2013) is not visible in the much longer dataset? Their study showed that the overshooting mode correspond to intense low-level reflectivity (and inferred larger and more numerous raindrop particles), which tend to occur more during the monsoon break period. Does the 14 km peak in break period (Fig. 6 MJO=1,2) correspond to that mode?

**With the new processing as suggested by previous comments from this reviewer, the 14 km peak has disappeared. However, we have added discussion on the presence of ETH greater than 15 km, which are only present during break conditions.**

12. The congestus mode in Kumar et al. (2013) is defined as ETH < 6.5 km, where in this study it is 8 km. That is not a small difference. 8 km is also significantly higher than the 0C ( 4.5-5.0 km) level where above which freezing and additional latent heating acceleration of vertical motion can occur.   Can you comment on why you choose a larger ETH value for congestus?

**We had originally defined the congestus and deep modes corresponding to the $\mu_1$ and $\mu_2$ derived from the bimodal Gaussian fits which resulted in larger values than what is typically considered congestus. In order to be more consistent with the past literature, we still base our classification of congestus and deep convection from the fits, but now place stricter criteria on what is classified as "congestus" versus "deep convective:"**
1.   **If the ETH distribution is bimodal (0.1 < A < 0.9) then mode 1 is the congestus mode and, mode 2 is the deep convective mode**
2.   **If the ETH distribution is unimodal (A < 0.1) and $\mu_2$ < 6.5 km then the single mode is the congestus mode, otherwise the single mode is the deep convective mode**
3.   **If the ETH distribution is unimodal (A > 0.9) and $\mu_1$ < 6.5 km then the single mode is the congestus mode, otherwise the single mode is the deep convective mode**

13.  Page 8 line 16-18, I thought A is the contribution of mode 1 (congestus), when A increase to 0.9 during active phases of the MJO under monsoon conditions, doesn't that mean most of the convection are congestus, as opposed to deep convection stated in this sentence? That is contrary to the statement of mostly widespread MCSs during active MJO and monsoon.  I think the issue here is using only echo-top height to indicate congestus vs. deep convection is too simplistic. The wide spread MCSs are likely associated with much larger but not as deep convective cells compare to more isolated but deep convection during break period.  One very important indicator for organized convection, i.e., the size of convective cells is ignored in this study.  Larger convective cells and larger updrafts (i.e. in MCSs) carry a majority of the mass flux as reported in both observational analyses (Kumar et al.  2015, Masunaga and Luo 2016) and high-resolution model results (Hagos et al.  2018).  The size of convective cells is just as important (if not more) as the depth of the convective cells. It can be easily quantified with the CPOL data and I think would be a useful quantity to investigate as functions of large-scale regimes along with the ETH analyses.

**We thank the reviewer for this helpful comment. We have taken care to more carefully define congestus and deep convection in response to this reviewer's major comment number 12. Doing this has improved the presentation of the results in this section and reduces confusion.**

[Figure]

We now discuss the convective areas as a function of large scale forcing, noting that there are generally lower convective areas in monsoon or active MJO conditions, which would be consistent with generally weaker convection being present during these conditions. We use this analysis, in combination with the quantitative MCS analysis

**suggested by the reviewer to show that monsoonal conditions are characterized by weaker, but more frequent MCSes.**

14. Page 8 line 29-30, I think this is an interesting and important finding. It would be useful to discuss what aspects of the large-scale conditions can help explain larger difference under active MJO (i.e. going back to Fig. 4-5, see major comment 10 about quantifying their relative environment profile difference between active/suppressed MJO, monsoon/break).

**Thank you. We have added a couple of sentences of discussion here on how the differences in the environmental profiles observed between MJO phases could be contributing to the differences in the ETHs seen:**
**"*Considering that, in Figures 4a and 5a show greater increases in equivalent potential temperature with height above the 5 km stable layer during MJO inactive conditions, this suggests that the midlevel thermodynamic profiles support greater inhibition of the convection in the deep convective mode when the active phase of the MJO is away from Australia.*"**

**With the new ETH processing and more careful definition of the modes, we also now observe that the congestus mode is more sensitive to the presence of the monsoon while the deep convective mode is more sensitive to the presence of the MJO**

15. Figure 9, other than the more obvious enhanced frequency of deep convective during daytime, the difference for the rest of the panels between MJO phases are difficult to see. Perhaps adding a difference panel would help.

**We attempted to add a difference panel to Figure 9, but this made the figure too busy to be readable. Given that the other reviewer commented that there were too many variables in the paper, we did not add a difference panel here.**

16. Page 9 line 26-29, why do you have to guess that the enhanced nighttime peak over the ocean during monsoon period is due to MJO? It is relatively easy to identify MCSs in CPOL data (e.g., Rowe et al. 2014 used a simple criteria of precipitation feature major axis length > 100 km to identify MCS in ground-based radar observations), why not actually quantify MCS frequency changes to better support your claim?

**We have quantified the radar coverage of MCSes using the methodology of Rowe and Houze (2014) for each scan as the reviewer suggested and have added the average number of MCSes in the radar domain per scan for a given large scale forcing regime and time of day in Table 1. The normalization was done to ensure that differences in the number of MCSes identified was not due to the differing lengths of time spent in each regime.**

The results in Table 1 clearly show that, during both an active MJO and active monsoon, that on average more MCSes are present in the radar domain during these conditions. On average, there is also increased presence of MCSes during the daytime compared to night time. Therefore, doing a quantitative analysis suggests that, in most conditions except active monsoon, there are more MCSes at night than during the day. Therefore, most of the conclusions we had before still hold, and where any changes had to be made the discussion was changed accordingly. We also now refer to the frequencies in Table 1 in addition to the already provided references to justify our claims of MCS frequency in Section 4.

17. Figures 10-11, given the strong diurnal cycle between land vs. oceanic area shown in Figures 8-9, did you separate land area and ocean area when calculating their ETH occurrence? I also suggest plotting Figures 10-11 in local time to make it easier for Readers.

We did not initially separate the diurnal cycle figures by land and ocean. At the suggestion of the other reviewer who was concerned about too many variables being presented in this section we have not added an extra figure separating the diurnal cycle by land and ocean.
Figures 10 and 11 are now plotted as a function of local time.

18. Page 11 line 13-14, which figure shows a peak of ETH around 5-6 km during break conditions? Figure 6 when MJO is away from Australia (phases 1-2) generally show a peak between 6-8 km, but drops to 4 km in phases 3-4.

We have modified this conclusion to be more consistent with the analysis in Figure 6, which has changed with the ETH reprocessing.

Minor Comments
1. Page 1 line 2, technically validation of convective processes in GCMs do not "require" such statistics, perhaps it's better to say "could benefit from" such statistics.

We have changed the wording in this sentence to the suggested wording from the reviewer.

2. Page 1 line 5, why does it have to be for a specific model? These observations can be useful for any large-scale model validations.

We were focusing on E3SM as for that model there is an undergoing development to have the model compute results at a 12 km resolution that is high enough such that both MCSes are resolves and the assumptions made in convective parameterizations may not apply. However, since we agree that this is really useful for any GCM, we have changed this sentence to state that this dataset is useful for the validation of convective

**processes in any GCM.**

3. Page 1 line 22, Jensen et al. (1994) stated the 100 W/m2 is solar forcing, not net radiative forcing, please clarify that in the statement.

**We have corrected this statement.**

4. Page 2 line 19, "...of of an intense..."

**We have removed the extra "of."**

5. Page 2 line 27, "...for one wet season in and found..." in what?

**We have completed this sentence**

6. Page 4 line 1, spell out the acronym "ACRF".

**Due to a recent change in the name of the ARM Climate Research Facility (ACRF) to ARM Facility, we have changed this to say "ARM Facility."**

7. Page 4 line 4-6, I believe the MTSAT data, at least the inferred channel spatial resolution should be 5 km, not 1 km. Also, the VISST technique uses all available geo-stationary satellite channels, including visible, water vapor, near IR, and IR channels to retrieve cloud properties during day time. For nighttime, a different technique SIST is used for the lack of visible channel.

**We now mention that the VISST technique uses these two techniques in this sentence. Also, according to the ARM archive, the resolution is 4 km, which we now mention here.**

8. Page 5 line 14, you mentioned "normalized frequency distribution", what is the unit of the shading in Figure 3? The large numbers appears to be just a count, if so it is not normalized frequency.

**We have removed the word "normalized" from this sentence.**

9. Page 5 line 31, do not use "cloud top height" and ETH interchangeably. You have not established in this study that CPOL ETH is equivalent to cloud top height.

**We have changed this phrase to say "echo top height."**

10. Figs. 3-4, why is the vertical scale different between the two figures?

**The vertical scales are all now the same between Figures 3 and 4.**

11. Page 7 line 12, spell out p.d.f. in section headings.

**We have corrected this to say "normalized frequency distributions" which is actually what is being plotted.**

12. Page 8 line 8, "In 7,..." do you mean in "Eq. (1)"?

**We meant to say Figure 7. It says so now.**

**References for Authors' Response**
Barnes, S. L., A technique for maximizing details in numerical weather-map analysis. Journal of Applied Meteorology. 3,4,: 396–409, doi:10.1175/1520-0450(1964)003<0396:ATFMDI>2.0.CO;2, 1964

Del Genio, A.D., Representing the Sensitivity of Convective Cloud Systems to Tropospheric Humidity in General Circulation Models, *Surv. Geophys*, 33: 637. https://doi.org/10.1007/s10712-011-9148-9, 2012.

My major criticism is that the description is confusing and hard to follow in parts of the statistical analysis (Sec. 4.1) and diurnal cycle (Sec. 4.2) sections

as described in the specific comments below. The confusion is due to 1) combination of too many variables to consider when trying to correlate interpretations stated in the text to results shown in selected figures: active vs inactive monsoon, MJO phase, day vs night, ocean vs land; and 2) the text is not always explicit in terms of which panel of a figure is being used to advance the argument. In consequence, I as the reader, have sometimes come to a different conclusion when interpreting the figures in question compared to the authors. I have also noted these instances below.

**We agree that there are quite a few variables that we are stratifying the ETHs and convective areas against and it can be quite overwhelming to the reader. However, the macrophysical properties of convection in Darwin are sensitive to numerous factors including the time of day, the phase of the MJO or the monsoon and even the surface characteristics. We strongly feel that it is important to show how the ETHs can vary as as a function of such characteristics.**

**Since we are unable to outright eliminate any independent variables from our analysis, we have taken some steps to simplify the presentation of the data to make it easier to interpret:**

1. **We now use a more rigorous definition of "congestus" versus "deep convection" that is based off of the threshold used by Kumar et al. (2013), who defined the boundary between congestus and deep convection to be 6.5 km. This, not only being more rigorous and consistent with past literature, also makes the results in Section 4.1 and 4.2 easier to read.**
2. **Every multipanel figure are now labelled by letter and each subpanel is now referred to explicitly.**
3. **The figures showing the ETH distributions as a function of time now just show the frequency distributions with time, reducing the number of figures by 1 and cleaning up clutter. This was done as there was little discussion and few conclusions to be made about how bimodality varies throughout the day.**

**Furthermore, the other reviewer requested a quantification of MCSes as well as an an analysis of convective areas in order to more adequately characterize the presence of deep convection than what was done in this manuscript. We have added such analysis to the statistical analysis section.**

A more modest critique concerns the spectrum width thresholding technique used to discriminate echo top height as opposed to a minimum reflectivity threshold. On the bright side, the method appears to work reasonably well. However, the end results is that there does not seem to be any real difference in the results when compared to simply applying a minimum reflectivity threshold (which is the traditional approach)
so I am left scratching my head when trying to understand the real advantage of the methodology.

**We do not use spectrum width in order to determine the echo top height, but rather, we use the texture of the radial velocity field. This is the standard deviation of the radial velocity of a 3 by 3 gate window surrounding a gate and not the standard deviation of the Doppler spectrum. Radars like CPOL that have a phase that varies randomly from pulse to pulse produce radial velocities that vary randomly gate-to-gate when there is no single scattering returned radiation.**

**The advantage of such an approach is that (1) the noise floor is automatically detected, (2) we can potentially be able to keep regions in cloud where Z is lower than 5 dBZ, which would be more frequent with 50 km range of CPOL. Finally, this methodology is immune to radar miscalibration and less sensitive to attenuation. This would give us, in theory, more representative of the true cloud top height. While we have removed the "novel" and "new" wording in the abstract, we now more explicitly list these advantages in the introduction and frame the discussion as a sensitivity test. We also have improved the explanation of the velocity texture based ETH retrieval technique so that it is easier to understand both how it works and its potential advantages.**

**While, for this particular dataset, we arrive at the null conclusion that the ETHs retrieved using the velocity texture methodology and using reflectivity are comparable. We feel that the inclusion of null results in a paper is something that should be done more to guide future research on what methodologies have been tried. This is not done enough in papers in our opinion, and given that this section is short, adding a null result does not significantly lengthen the paper.**

Specific Comments 1. P. 4, line 12; please define gate spacing and resolution

**We have added the 300 m resolution and gate spacing to this section.**

2. P. 5, line 7: What is the spatial resolution of the satellite data? If it's less than radar resolution it's not clear what a relative comparison tells us regarding the performance of the radar-based ETH algorithm.

**The spatial resolution of the satellite data at 4 km, so interpolation of the satellite data to the radar's grid was needed.**

**In response to the other reviewer, we have expanded the analysis in this section to 4 years of MTSAT data from 2006 to 2010, using version 4 of the VISST product. While we acknowledge that the coarser resolution of the satellite introduces uncertainties into the satellite retrieval, over time scales of years the relative seasonal variability in cloud top heights should be captured.**

3. P. 5, line 15: similar to previous comment - to understand the differences in cpol vs satellite – what is satellite brightness temp keying off of – what depth of cloud is considered?

**The VISST technique uses both the solar and infrared channels at multiple wavelengths to retrieve the cloud properties. According to Cheng et al. (2010), the retrieval is keying off of a height "somewhere**

below physical cloud top," but they do not quantify exactly where. We have chosen not to attempt to quantify this since this is extremely difficult to do.

4. P. 5, line 17: cc of 0.49 is not very good

**This, while weak, is a statistically significant correlation according to a chi-squared test. The mention of the statistical test has been added to this sentence.**

5. P. 5, line 20: this statement assumes the satellite is capturing the variability …

**Over timescales of seasons and spatial scales of hundreds of kilometers, we would fully expect this to be the case.**

6. Fig. 3 please state in the caption what the color shading represents

**We now state that the shading represents the number of counts.**

7. P 6. Lines 25-30: In references to Figs 4-5, seems like the big differences are between monsoon phase instead of MJO phase?

**For the winds, this is the case. We have noted that there is little difference between the wind speeds between active and inactive MJO conditions.**

8. P. 7, line 7 – There are several other older references
that show this behavior: Cifelli and Rutledge 1994 (JAS); 1998 (QJRMS)

**We have added these references to this sentence.**

9. P. 7, line 27-28: some hint of trade wind layer in MJO=3 for break (Fig. 6)?

**We have now changed this sentence to acknowledge that we could be sensing some of the trade wind layer during break conditions, but we ultimately need a radar that is sensitive to cloud particles to characterize it:**

**"***Also, some evidence of the trade wind mode is visible in Figures 6a-h. However, since the 2 km modes in Johnson et al. (1999) and Kumar et al. (2013) were observed using measurements with a cloud radar that would be more sensitive to liquid cloud droplets than CPOL, more sufficient quantification of this mode would require a radar with a lower minimum discernable signal than CPOL.***"**

10. P. 8, line 8: This is a minor point but it should be noted that the heights of the different modes that are stated here are approximate. For example, in Fig. 7c the height of mode 2 does not appear to actually reach 15 km

...

**We have added the word "approximately" before the quantities in this sentence, and fixed the faulty reference to Figure 7.**

11. P. 8, line 12-18: there is some confusion looking at Fig 7. My read of the red line (A=congestus) in MJO phases 4-7 is ~0.05 – 0.6 for break (Fig 7b) – not 0.8-0.5 as described - and ~0.1-0.4 for monsoon (Fig. 7d) – not > 0.9 as described. Also, the statement on line 14 about unimodality is confusing:

Fig 7b,d show that there is a significant contribution from the congestus mode in break conditions while the MJO is over AU (Fig. 7b, MJO phases 6,7). Similar in monsoon conditions for MJO phase 6 – see Fig 7d. I think the confusion noted above could be avoided by stating more clearly which features in specific figure panels are being referred to.

**We have made a more rigorous definition of "congestus" versus "deep" convective modes that makes this section easier to understand. We**

**determine whether the modes present are congestus or deep convection based on the average location of the modes:**

1.  **If the ETH distribution is bimodal (0.1 < A < 0.9) then mode 1 is the congestus mode and, mode 2 is the deep convective mode**
2.  **If the ETH distribution is unimodal (A < 0.1) and $\mu_2$ < 6.5 km then the single mode is the congestus mode, otherwise the single mode is the deep convective mode**
3.  **If the ETH distribution is unimodal (A > 0.9) and $\mu_1$ < 6.5 km then the single mode is the congestus mode, otherwise the single mode is the deep convective mode**

**The 6.5 km threshold was chosen based off of Kumar et al. (2013).**

**We agree, the statement about the unimodality was confusing. We removed it.**

12. Fig. 8 – please state in the caption and the figure that this is for break conditions

**We have added this information to the caption.**

13. P. 9 -please call out panels explicitly in reference to Figs. 8-9

**Labels have been added to these panels and now the discussion refers to each relevant panel explicitly.**

14. The discussion jumps to Fig 10 before discussing Fig. 9

**This is no longer the case.**

15. P. 9, line 24: which panel of Fig 10? My read of comparing Fig 10 a and Fig 10b is that during the day there is a higher frequency of deep convection when the MJO is over AU (assume that includes Tiwi islands as well) compared to when MJO is elsewhere.

**We have added a reference to Figure 10d.**

16. P. 9, 25-26: I don't understand the point about what is being extended in this study vs previous work.

**We agree, this sentence was quite confusing. We have rephrased this sentence from:**
"*Figure 10d shows a greater frequency of deep convective ETHs over the Tiwi Islands when the MJO is inactive over Australia during the day, which is consistent with increased rainfall over this region.*"

17. P. 10 lines 11-12 – where do the number of days come from?

**The number of days comes from the sounding classification in Section 3.3**

**References:**

Chang, F.-L., P. Minnis, J. K. Ayers, M. J. McGill, R. Palikonda, D. A. Spangenberg, W. L. Smith Jr., and C. R. Yost (2010), Evaluation of satellite-based upper troposphere cloud top height retrievals in multilayer cloud conditions during TC4, *J. Geophys. Res.*, 115, D00J05, doi: 10.1029/2009JD013305.

[revised manuscript text omitted]

---

## Author Response (AR3)

Review: ACP-2018-408-R1

Title: A 17 year climatology of the macrophysical properties of convection in Darwin Recommendation: Minor Revision.

**Summary**

The authors have done extensive revision to address many of my comments. I particularly commend the authors to extend their analysis beyond just using the vertical height of convection, but also look into the horizontal dimension of convection, including identifying MCSs explicitly. The added results on convective cell sizes and MCSs nicely complement the ETH analysis of the study. As a result, I think the paper has improved significantly compared to the original manuscript.

There are a few places with some relatively minor remaining issues that should be revised/clarified before the paper is published. Please see my comments in orange below. I recommend minor revision.

We thank the reviewer for their time and effort in reviewing the revised manuscript. We have made changes in accordance to the comments below. In addition, we have added a reference to the calibration technique that we used to calibrate the reflectivity values (Louf et al. 2018, JTECH) that was just accepted for publication. We have embedded responses to each individual comment in bold below. Furthermore, it was found that Figure 6 (now figure 7) was not consistent with the revised processing, so that figure was regenerated which did not change the conclusions of the paper.

**Comments**

Page 6 line 9-12, actually the daytime satellite retrieval algorithm is referred to as VISST, and the nighttime algorithm is referred to as SIST: https://cloudsway2.larc.nasa.gov (click the VISST/SIST link).

**We have changed this sentence to have the correct names for each technique.**

7. There is also the issue of daytime vs. nighttime retrieval differences in the MTSAT data. ...

We had not originally done these separations. We went ahead and separated Figure 3c by retrievals made in the daytime and excluded twilight and placed them in the figures below. Our results are insensitive to the time of day.

Thank you for attempting to separately compare CPOL and MTSAT echo/cloud-top height retrievals between day vs. night. It's good to know that the results are insensitive to the two different algorithms, suggesting at least that the VISST/SIST provide statistically consistent convective cloud-top height retrievals. More importantly, comparing your original Figure 3 with the new one where the CPOL ETH is now defined as the lowest precipitating convective echo-top, the mean differences as well as the spread is now significantly larger (i.e., CPOL ETH is now significantly lower than MTSAT for cloud-tops above 7.5 km. That means the occurrence of multi-layer clouds above precipitating convective cells are indeed quite frequent, obviously the passive satellite only "sees" the highest layer cloud tops for these optically thick clouds, while the CPOL could detect distinct layers. I think this is a useful result to point out, perhaps you could add a couple sentences describing the difference when comparing CPOL lowest layer ETH vs. max ETH.

We agree with the reviewer that this is an interesting conclusion to highlight that shows the importance of using scanning radars over satellites to estimate the heights of convective systems and thank them for their suggestion. We have added text at the end of this section to discuss this difference:

"Because of this definition, the fact that, when the VISST CTH is greater than 10 km, the median CPOL ETH is 4 to 5 km lower than the VISST retrieved CTH in Figure 3 shows that there are commonly multiple layers of cloud present. This therefore shows that CPOL is able to detect the presence of multiple cloud layers while the VISST technique can only detect the highest cloud layer and shows an advantage of using the CPOL ETHs over the VISST CTHs."

Figure 4. I don't think the temperature profile (Fig. 4b) provides much useful information. Similarly, dew point temperature and specific humidity is also somewhat duplicative. You also did not specifically discuss these quantities in the text. Relative humidity is a more useful quantity that differentiates probability of transitioning to deep convection. I suggest you could simplify Fig. 4, 5 with just 4 panels, theta E, RH, U, V.

**We have reduced Figures 4 and 5 to the suggested panels and modified the discussion accordingly.**

Page 7 line 33-34, "This shows that there are a greater number of cases with westerly flow advecting moisture from the Indian Ocean when the MJO is active over Australia." I think the 95 th percentile U wind being larger when MJO is over Australia in Fig. 4e, 5e only means the tail of the zonal wind is stronger. That does not necessarily mean moisture advection is larger. You could have stronger zonal wind with drier air. What would support your claim is to compare moisture flux (U \* qv) profiles.

Figure R1. Calculated moisture fluxes from rawinsonde in break conditions (top) and active monsoon conditions (bottom). Blue lines represent active MJO conditions, and red lines represent inactive MJO conditions. Solid lines are medians, dashed lines are 5th/95 percentiles.

We thank the reviewer for their helpful suggestion. We have calculated the moisture fluxes from the rawinsonde data by using U\*qv and plotted the statistics of the profiles in Figure R1. The medians and 95th percentiles show that during active MJO conditions, the moisture fluxes in the lowest 1 km of the atmosphere are greater, indicating that more surface moisture is being advected into the area creating a more favorable environment for convective initiation. In addition, there are greater moisture fluxes during active monsoon conditions than during break conditions. Figure R1 is now added as Figure 6 in the paper with the relevant discussion added showing how the environmental conditions favor the development of convection during an active MJO and active monsoon.

Figure 7d, what concerns me is the complete reverse between 100% deep convection in MJO phase 1 to 100% congestus in MJO phase 2. Is it reasonable to believe that: 1) 100% of convective clouds during when MJO is away from Australia are all deep convection (or congestus)? 2) All of the deep convective clouds suddenly all changed to congestus in the next MJO phase? You also ignore this figure in the text when discussing Fig. 7 on page 10. I don't think I understand what u1 and u2 mean, and how are the fractional contribution of modes in Fig. 7b,d are calculated.

Given the locations of the deep convective and congestus modes in Figure 7, this switch is likely due to the location of the single mode in Figure 7c at MJO phase 1 being close to the threshold of 6.5 km (7.5 km) that was used to separate deep convection and congestus, while it is 6.5 km during MJO phase 2. We add text showing that this is a potential limitation of discriminating congestus with deep convection using ETH alone along with this explanation in this section. We have also added additional text in this section clarifying how  $\mu_1$ ,  $\mu_2$ , and A are used to determine the locations and contributions of the deep convective and congestus modes.

I find the new Figure 8 quite interesting. Thank you for adding this analysis. The result suggests that when MJO is over Australia, monsoon/break has no effect for the population of convective cells. But when MJO is away from Australia, monsoon periods have narrower distribution of convective cell sizes, and also less frequent large convective cells than break period. I think this could be one of the highlighted results of the paper.

**We agree that this is a very interesting conclusion of this study. We have added text to the conclusions, abstract, and the paragraph discussion Figure 8 highlighting this conclusion.**

Table 1 caption is misleading, the numbers in the table are frequencies, not "average number of MCSs". The caption also has some missing information: "... using the criteria of (?)".

**We have fixed this table caption.**

Page 11 line 15-16, Table 1 only provides MCS frequencies separately for break/monsoon, and MJO away/over Australia (unless I misunderstood). But it does not provide frequencies during break+MJO away from Australia, break+MJO over Australia, monsoon+MJO away, monsoon+MJO over. So it is inconsistent with the discussions in Fig. 7 in this paragraph, which combines the monsoon and MJO conditions. How about adding the MCS frequency calculations for these combined conditions in the table to align better with the 4 key large-scale conditions of the paper?

**We had added these entries to Table 1 for the combined conditions. Thankfully, this does not change our discussion much, but we agree with the reviewer that it is valuable to show both the data stratified for the 4 categories separately as well as for the combined conditions.**

Page 13 line 12-14, "Since MCSs are larger ..., the reduced frequency of MCSs in the inactive phase of the MJO in Table 1." The last part of the sentence seems to have some missing words.

**We have finished this sentence.**

1Argonne National Laboratory, 9700 Cass Ave., Lemont, IL, USA

[revised manuscript text omitted]